**Registered report**

# No evidence for decision fatigue using large-scale field data from healthcare

David Andersson[1], Malou Lindberg[2], Gustav Tinghög [1,3] & Emil Persson [1] ✉

Decision fatigue is the idea that making decisions is mentally demanding and eventually leads to deteriorated decision quality. Many studies report results that appear consistent with decision fatigue. However, most of this evidence comes from observed sequential patterns using retrospective designs, without preregistration or external validation and with low precision in how decision fatigue is operationalized. Here we conducted an empirical test of decision fatigue using large-scale, high-resolution data on healthcare professionals' medical judgments at a national telephone triage and medical advice service. This is a suitable setting for testing decision fatigue because the work is both hard and repetitive, yet qualified, and the variation in scheduling produced a setting where level of fatigue could be regarded as near random for some segments of the data. We hypothesized increased use of heuristics, more specifically convergence toward personal defaults in case judgments, and higher assigned urgency ratings with fatigue. We tested these hypotheses using one-sided Bayes Factors computed from underlying Bayesian generalized mixed models with random intercepts. The results consistently showed relative support for the statistical null hypothesis of no difference in decision-making depending on fatigue ($BF_{0+} > 22$ for all main tests). We thus found no evidence for decision fatigue. Whereas these results don't preclude the existence of a weaker or more nuanced version of decision fatigue or more context-specific effects, they cast serious doubt on the empirical relevance of decision fatigue as a domain general effect for sequential decisions in healthcare and elsewhere.

Decision fatigue refers to the idea that making decisions is mentally demanding and therefore depleting, which in turn affects the content and quality of subsequent decisions[1–4]. It can be seen as a specific instance of the more general notion mental fatigue[5–8], with overlapping proximal causes but the emphasis on decision-making. Decision fatigue has intuitive appeal and is widely applicable to everyday decision-making. It is also relevant in expert judgment with high stakes, such as law, medicine and finance; because these are decision environments typically characterized by handling of complex back-to-back cases under pressure to work fast. Thus, establishing the factual existence of decision fatigue, and quantifying its effect in various domains, is important. Many previous studies presented intriguing empirical patterns that are in principle consistent with decision fatigue[2,9–18]. There are also calls for policy to mitigate these effects, most notably in healthcare[19–21], thus taking existing results at face value. However, most studies to date have used retrospective observational designs, without pre-registration or external validation, and there is generally a lack of precision

in how decision fatigue is operationalized. Direction of effects "predicted" by decision fatigue are seldom ex-ante obvious but seem easy to rationalize ex-post. Taken together, this means that there is a lack of good evidence for (or against) decision fatigue as a practically meaningful concept. In this Registered Report, we attempt to close this gap, using large-scale, high-resolution field data from healthcare.

The literature on decision fatigue spans multiple disciplines, with a recent concentration of papers in medicine. A landmark study on judicial decision-making found that judges were much more likely to give parole to prisoners seen in the morning or after a food break[2]. Noting that case ordering was plausibly random, the pattern was interpreted as indicative of decision fatigue, conceptualized as an increased tendency to use decision simplification strategies (e.g., rule in favor of the status quo and risk minimizing) when mentally depleted following a series of hard decisions. These results spurred much debate[22–25], primarily arguing that the magnitude of the effect was likely overestimated. There was a conceptual replication[26]

[1]Department of Management and Engineering, Division of Economics, Linköping University, Linköping, Sweden. [2]Department of Health, Medicine and Caring Sciences (HMV), Linköping University, Linköping, Sweden. [3]Department of Health, Medicine and Caring Sciences (HMV), The National Center for Priority Setting in Health Care, Linköping University, Linköping, Sweden. ✉e-mail: emil.persson@liu.se

(confirming but indicating a weaker effect, to date not published), and subsequent studies have presented evidence both in favor[27] and against[28,29] decision fatigue as a relevant factor in judicial decision-making. In finance, decision fatigue has been recognized as deteriorated forecasting accuracy and increased use of heuristics in decision-making (e.g., issuing rounded forecasts) over the course of a workday, or following a series of hard decisions[15,30–32].

In medicine, decision fatigue has been attributed to a number of outcomes related to sequential decision-making, often concerning the timing or order of patient appointments. Prominent examples include elevated prescription rates of antibiotics for patients seen later in the day[10], more conservative surgery recommendations as case ordering increased or just before lunch[13], increased likelihood of physicians prescribing painkillers (opioids) later in the workday[16,33,34], and elevated urgency ratings in triage or telephone assessment by healthcare professionals as time or workload measured since the last rest break increased[14]. A range of studies presented similar results and interpretations for other contexts in healthcare, including cancer screening and compliance with handwashing protocols[11,17,35–42]. A small number of studies explicitly argued that they found evidence against decision fatigue[43,44].

The main problem in the existing literature on decision fatigue is inadequate control of analytic flexibility (researcher degrees of freedom). Analytic flexibility refers to the many decisions that must be taken when collecting, cleaning, analyzing, and interpreting data[45,46]. If this is not carefully accounted for in design or analysis (e.g., ex-ante restricted via preregistration[47–50]), conclusions will be invalid. The underlying problem here is that a single scientific hypothesis often corresponds to a range of possible statistical hypotheses, meaning that there are many different ways (many different realizations of data) to claim evidence for an overarching scientific hypothesis[46]. For the case of decision fatigue, this is a real concern because evidentiary claims in this literature are based primarily on observed sequential patterns in noisy field data without strong theoretical guidance on how to operationalize cause or consequence of fatigue. Lack of focused operationalization means that interpretive researcher degrees of freedom abound in this literature. Since few studies have attempted to restrict analytic flexibility via preregistration or out-of-sample validation, cumulative evidence for (or against) decision fatigue is still weak, despite the many studies reporting such effects.

The goal of this study is to conduct a focused and strong empirical test of decision fatigue as a practically relevant concept for understanding repeated or effortful decision-making. To this end, we collect and analyze healthcare professionals' (specialized nurses) medical judgment in a large number of cases. The context is a national telephone triage and medical advice service in Sweden, 1177 direct. This is a suitable setting for observing and testing decision fatigue, because it offers data with a high level of granularity, with details on timing, length and sequence of cases (phone calls). Medical judgments are made on a five-level monotonically increasing scale of urgency, and we can track for each case whether other co-workers or external medical experts were also consulted. Moreover, since this is both hard and repetitive, yet qualified, work, it is plausible that workers feel tired and fatigued as the workday progresses. Thus, if there are relevant downstream behavioral correlates of fatigue, as the concept decision fatigue presupposes, this should be a good setting to observe such effects.

A worry in many of the published studies on decision fatigue is that case ordering or timing is determined in advance (scheduling) and as such is possibly influenced by unobservable factors correlated with outcomes. This can produce a false pattern that looks like decision fatigue but really is due to something else. In our setting this is not a problem because case assignment is essentially random at a given time point, among workers, which rules out deliberate sorting of cases (scheduling) based on prior knowledge of patient type or characteristics. There is also substantial variation in timing and length of workdays within individuals. This means that for some segments of the data, cumulative effort (which is plausibly linked to fatigue) at a given time point could be regarded as randomly assigned to each worker. Part of our design is thus quasi-experimental.

Decision fatigue has a plausible conceptual basis, consisting of three main building blocks. First, qualified decision-making in hard cases often involves cognitively demanding reasoning, which requires mental effort. This includes retrieving and interpreting information, engaging in tradeoff thinking by weighing the pros and cons of different courses of action and performing mental simulations of potential outcomes[51,52]. Second, prolonged mental effort is associated with aversive feelings of fatigue, boredom, and sometimes frustration[5,6,53–56]. An active literature in experimental cognitive psychology investigates the proximate causes of this relationship, and an emerging hypothesis is that the phenomenology of mental effort reflects the opportunity costs of controlled cognition[5,6,53,57]. As such, effort can be conceptualized as a cost, and subject to rational allocation based on expected costs and benefits[6,58,59]. Third, mental cost of effort influences decision-making. Here, the primary pathway goes via reduced willingness to exert effort[5–7,53,54,60] (due to increased cost) and consequently increased reliance on heuristics and simplification strategies when making decisions, and increased preference for easy, simple options. Reduced capacity for self-regulation when mentally 'depleted' has also been used as an outcome associated with decision fatigue[1,4]. Taken together, this conceptualization is consistent with the idea that decision fatigue can be seen as a specific instance of mental fatigue, which presupposes that prolonged cognitive activity (in this case primarily decision-making and hard thinking) may lead to aversive feelings and tiredness and subsequently a willingness to disengage[5–8].

Building on these underlying ideas, we derive testable hypotheses for how fatigue (cumulative mental effort) affects decisions in our setting using rational inattention theory from economics[61–63]. This framework is based on the notion that on the one hand, information is beneficial because it improves decisions when there is uncertainty (because uncertainty can be resolved and better actions chosen), but on the other hand, information is costly to acquire and process because it involves cognitively demanding reasoning and mental effort. As such, there is a fundamental tradeoff between saving mental effort and acquiring and processing more information to improve decisions[61,62]. Rational inattention theory characterizes the optimal information-processing strategy under these conditions, i.e., the optimal allocation of mental effort for the purpose of acquiring and processing information. The decision maker behaves as a Bayesian expected utility maximizer. She seeks to balance upfront the expected mental costs of acquiring and processing more information with the anticipated benefits of updating her prior beliefs about the state of the world, in our case her beliefs about the type, state, and severity of patients' medical problems. The optimal information-processing strategy in a setting with discrete choices, as in our case, results in probabilistic choice that follows a logit formula, reflecting both the utility of the best action under perfect information (i.e., the 'true payoff') and the decision maker's prior belief, as well as the cost of information[61].

In our setting, we think of information processing broadly as the predominant activity during consultancies with patients; listening, asking questions, assessing responses and trying to map them to indicators of known medical problems, ultimately to assign an appropriate level of urgency for onward contact with the healthcare system. Advice and recommendations for self-care are also provided. These activities are plausibly more cognitively demanding when tired, for example having worked consecutively for several hours, at the end of a long shift, or during night shifts. We incorporate decision fatigue in the rational inattention framework as an exogenous increase in the cost of information, which links back to the established idea that processing information is associated with costs of mental effort[51,56,58]. The predicted effect of increased information costs in this model is that the decision-maker plans to acquire and process less information, and consequently, choices become more sensitive to prior beliefs[61]. Whereas prior beliefs are, naturally, not observable in our data, we can plausibly think of them as an internal probability distribution over patient types and thus they should be correlated with each specialized nurse's unconditional choice frequencies, i.e., how often they assign the

highest level of urgency, second highest, and so on. This leads to our first hypothesis.

*Hypothesis 1.* With higher levels of fatigue, choices are more likely in line with personal defaults. We measure personal defaults as the most common choice for each worker (specialized nurses).

Our second hypothesis is based on the same underlying prediction, but instead of looking for moves toward personal defaults for each individual, we look for shifts in the aggregate distribution of choices. We hypothesize that, overall, urgency ratings will be higher with higher levels of fatigue. Increased use of the 1–2 highest scale points in particular corresponds to plausible information-processing strategies, for example in situations where a severe condition seems unlikely but cannot be ruled out without further inquiry (collecting and processing more information). Such a strategy would mean that less effort is needed while also in some way choosing the safest option for both the patient and the specialized nurse handling the case. This is also in line with the previous literature, where increased preference for easy, safe, conservative, options is often rationalized as decision fatigue[10,13,14].

*Hypothesis 2.* With higher levels of fatigue, urgency ratings are higher on average.

Confirmatory assessment of decision fatigue relies on tests of these two hypotheses. We proxy for fatigue using an observational correlate of cumulative mental effort, focusing on time on shift in a quasi-experimental comparison during a time window when shifts overlap (morning vs. afternoon) and intense call sequences around breaks, both lunch and shorter. Based on exploratory analysis of pilot data, we decided to disregard measures linked to time of day that were previously popular in the literature, because we saw a high risk of confounding due to time patterns in patient composition, e.g., severity of medical problem. For each hypothesis, we quantify relative strength of evidence for an effect in the hypothesized direction vs. a point null hypothesis of no effect, using one-sided Bayes Factor hypothesis testing and BF > 10 as a criterion for conclusive evidence. This means that outcomes in line with our predictions would be generally interpreted as favouring decision fatigue, and, conversely, outcomes in the opposite direction or in favor of the null (indicating no difference) would be interpreted as evidence against decision fatigue. See Table 1 (Design Table) for an overview of planned analyses, evaluation, and design.

### Preregistered exploratory questions

We explore effects related to call duration and proportion of calls resulting in external consultation. Observed shifts in these outcomes can plausibly be rationalized as decision fatigue, but it is not obvious what direction of effects to expect ex-ante; e.g., call duration could be shorter because workers retrieve less information on the margin, or longer because workers are more tired and therefore need more time to retrieve each "unit" of information.

## Methods

### Protocol registration

The Stage 1 protocol for this Registered Report was accepted in principle on May 7, 2024. The protocol, as accepted by the journal, can be found at https://doi.org/10.6084/m9.figshare.25817575.v1.

### Ethics information

The research complies with all relevant ethical regulations. The study protocol was approved by the Swedish Ethical Review Authority (ref 2021-03639), including a waiver of informed consent for the registry data used in the study.

### Pilot data

We collected and explored a large pilot data set via retrospective observational analysis of $n = 203,100$ medical judgments made by $k = 200$ healthcare professionals (specialized nurses). The setting was a national telephone triage and medical advice service operated by regional healthcare providers in Sweden (1177 direct). We used this data set to find the most suitable ways of testing hypotheses about decision fatigue. We searched for good comparison points in the data, where the presupposed influence of fatigue on

decisions would not be confounded by other factors (e.g., temporal flow of different patient types). Two such comparison points emerged: a) A temporally confined overlap between morning and afternoon shifts, where level of fatigue was plausibly as-if randomly assigned (due to variation in scheduling) (Supplementary Fig. 1), and b) before vs. after breaks for intense call sequences.

Using these data, we tested for influence of fatigue on urgency ratings and likelihood of choosing in line with personal defaults (our main hypotheses, see Introduction) with one-sided Bayes Factor (BF) hypothesis testing and furthermore calculated two-sided Bayesian 95% credible intervals for these outcomes as well as for call duration and external consultancy. These measures were computed from underlying Bayesian generalized mixed effects models with random intercepts for nurses.

Results indicated relative support for the statistical null hypothesis of no difference in outcome depending on fatigue ($BF_{0+} > 45$ for all tests) and the 95% credible intervals generally had little mass on parameter estimates of sufficient magnitude in the expected direction. We also explored time of day as an alternative proxy for fatigue (which is a common approach in the literature) and found a pattern that could be interpreted as decision fatigue, although using time of day in this way is problematic because it risks confounding due to other temporal patterns in the data. Taken together, analysis of the pilot data provided no good evidence for decision fatigue. See the Stage 1 protocol and associated supplementary material (available at https://doi.org/10.6084/m9.figshare.25817575.v1)[64] for more details on the pilot data collection and analyses.

### Design

**Summary of approach**. Here we conduct a direct confirmatory test of the decision fatigue hypothesis in a new sample from the same setting as the pilot data (planned $n = 270,000$, $k > 200$; achieved $n = 231,076$, $k = 174$), using the most relevant analysis paths followed in the explorative analysis of that data. The achieved sample was smaller than expected likely because the pilot data period was unusually busy due to the ongoing COVID-19 pandemic (we used a linear extrapolation from the pilot data sample size to compute expected sample size for a fix target data time period). The design is retrospective observational but planned in advance, and with quasi-experimental components just like in the analysis of the pilot data. We did not have access to the new data before the Stage 1 protocol was accepted in principle and date stamped.

**Setting**. The setting for our study is a national telephone triage and medical advice service operated by regional healthcare providers in Sweden (1177 direct). There are 21 regional providers (units) of this service. We selected three of them based on our prior familiarity with their organizational setting and work environment. Around 200 specialist nurses work in these three units (in total) and they receive ca 1200 patient phone calls each day (October 2020). Patients can call for advice on any medical problem they have. For each call, the assigned nurse makes a medical judgment by assigning an urgency rating (5-point increasing scale) for how urgently the patient needs in-person medical attention. This will be our main outcome variable. We can also track for each case whether the nurse consulted with a co-worker or an external medical expert.

Patients are assigned to nurses in a pseudorandom order as calls come in. This is good because it prevents confounding due to active sorting of cases, e.g., more urgent seen first. There is also substantial variation in scheduling within individual nurses. Morning, afternoon, and evening shifts come in different configurations, with staggered start times and varying lengths. They are staffed primarily from the same pool of staff on a rolling basis, meaning that each nurse works both morning and afternoon/evening shifts. Night shifts are different and mainly staffed recurrently with the same individuals. Figure 1 shows the distribution of first and last calls each shift.

Shifts vary in length. The majority of them are between six and eight hours long, except night shifts which are around ten hours. There are also mixed shifts involving both working with patient calls and performing other

**Table 1 | Design Table**

| Question | Hypothesis | Sampling plan (e.g., power analysis) | Analysis Plan | Interpretation given to different outcomes |
|---|---|---|---|---|
| Do specialist nurses at a higher level of fatigue more often choose in line with personal defaults? | A higher proportion of choices will be the same as personal defaults when fatigue is high compared to low. | Test in two most suitable[a] samples: a) AM-PM shift overlap; b) Before vs. after breaks. Total sample size was determined based on suitability of available data and ensuring sufficient probability of detecting relevant effects and low probability of generating misleading evidence. We used a fixed-n approach to simulate power for Bayes Factor analysis for an approximated paired design for effect of fatigue. For an assumed positive effect $\sim N(0.2, 1)$ we obtained 99.9% correct classification using BF > 10 as criterion for conclusive evidence. For an assumed null effect $\sim N(0, 1)$ we obtained 83.9% correct classification and < 1% misclassified. | In each sample (overlap and breaks), fit a Bayesian generalized mixed effects model for binary DV (here *Default*) on main independent variable capturing high vs. low fatigue (e.g., *Morning* in overlap sample), with random intercept for nurses. In overlap sample, additionally include time indicator variables as fixed factors. We use weakly informative priors (see Methods section for details). Compute one-sided Bayes Factors $BF_{+0}$ to quantify evidence for effects in the predicted direction ($H_+$) vs. a point null hypothesis of no effect ($H_0$). | Interpret outcomes in line with prediction as in favor of decision fatigue; interpret outcomes in the opposite direction or in favor of the null (indicating no effect from fatigue) as against decision fatigue. Formally, $BF_{+0}$ computed and interpreted directly for each test as magnitude of relative evidence for $H_+$ vs. $H_0$. Hypothesis summary assessment based on two such Bayesian hypothesis tests (i.e., one in each sample a–b): Hypothesis is *supported* if $BF_{+0} > 10$ for at least one of the two tests; hypothesis is *not supported* if both BFs satisfy $BF_{+0} < 1/5$ AND at least one of them $BF_{+0} < 1/10$; otherwise evidence is *inconclusive*. |
| Do specialist nurses at a higher level of fatigue choose higher urgency ratings on average? | Urgency ratings will be higher on average when fatigue is high compared to low. | Test in two most suitable[a] samples: a) AM-PM shift overlap; b) Before vs. after breaks. Total sample size was determined based on suitability of available data and ensuring sufficient probability of detecting relevant effects and low probability of generating misleading evidence. We used a fixed-n approach to simulate power for Bayes Factor analysis for an approximated paired design for effect of fatigue. For an assumed positive effect $\sim N(0.2, 1)$ we obtained 99.9% correct classification using BF > 10 as criterion for conclusive evidence. For an assumed null effect $\sim N(0, 1)$ we obtained 83.9% correct classification and < 1% misclassified. | In each sample (overlap and breaks), fit a Bayesian generalized mixed effects model for continuous DV (here *Urgency*) on main independent variable capturing high vs. low fatigue (e.g., *Morning* in overlap sample), with random intercepts for nurses. In overlap sample, additionally include time indicator variables as fixed factors. We use weakly informative priors (see Methods section for details). Compute one-sided Bayes Factors $BF_{+0}$ to quantify evidence for effects in the predicted direction ($H_+$) vs. a point null hypothesis of no effect ($H_0$). | Interpret outcomes in line with prediction as in favor of decision fatigue; interpret outcomes in the opposite direction or in favor of the null (indicating no effect from fatigue) as against decision fatigue. Formally, $BF_{+0}$ computed and interpreted directly for each test as magnitude of relative evidence for $H_+$ vs. $H_0$. Hypothesis summary assessment based on two such Bayesian hypothesis tests (i.e., one in each sample a–b): Hypothesis is *supported* if $BF_{+0} > 10$ for at least one of the two tests; hypothesis is *not supported* if both BFs satisfy $BF_{+0} < 1/5$ AND at least one of them $BF_{+0} < 1/10$; otherwise evidence is *inconclusive*. |

[a]Samples are selected for best possible comparison based on explorative analyses of the pilot data; "best" means external conditions for both groups are as similar as possible except level of fatigue, which should be as different as possible. See Sampling plan subsection for details.

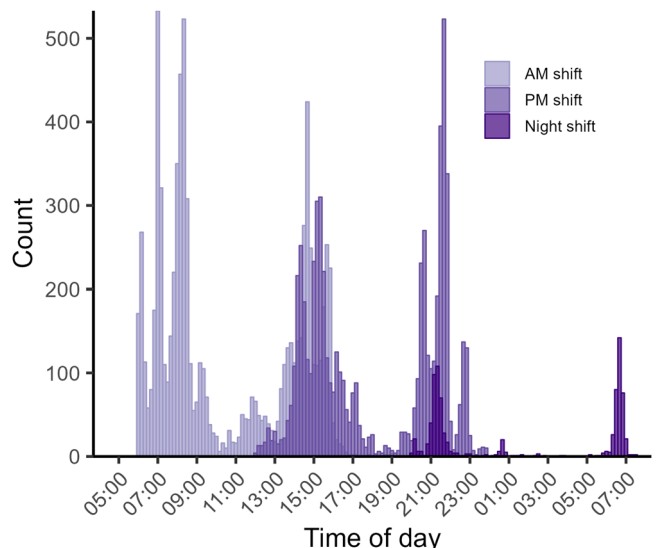

**Fig. 1 | First and last call each shift.** Substantial overlap in the afternoon between morning and afternoon shifts gives a quasi-experimental setting. AM shift indicated in light purple, PM shift in purple, and Night shift in dark purple [*n* = 15,810].

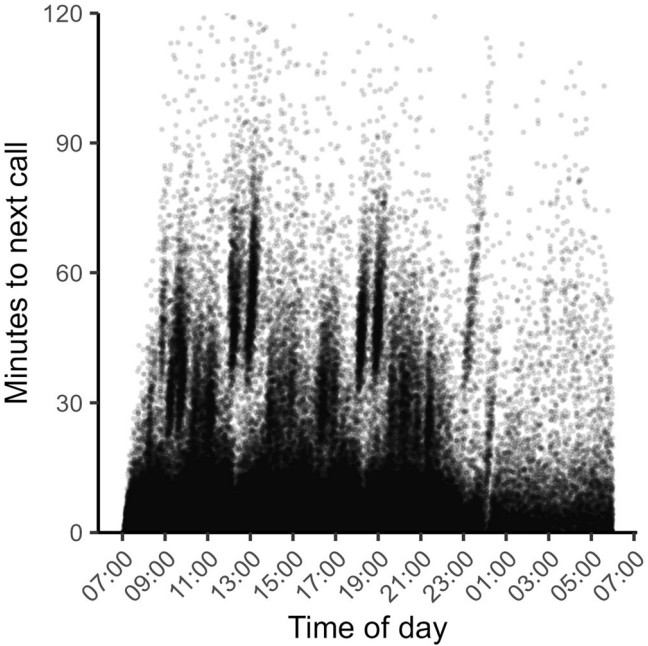

**Fig. 2 | Timing and duration of breaks.** Each point in the figure represents an individual call indexed by the time it ended (x-axis) and duration in minutes to next call for the same nurse on the same day (y-axis) [*n* = 145,991].

tasks, e.g., administrative. There is usually a high level of intensity (during normal shifts). On average, 21.0 (SD = 7.0) patient calls are handled during a typical 6–8 h shift. There is a 30 min lunch break and 1–2 shorter breaks of 15 min each. Calls are handled consecutively with little time in between; on average 4.8 (SD = 4.5) min between calls, including time for case documentation but excluding breaks >15 min. Most calls are short, median 6.6 (IQR = 6.6) min. (Data source: target data.) Corresponding statistics in the pilot data: On average, 25.3 (SD = 8.5) patient calls are handled during a typical 6–8 h shift. There is a 30 min lunch break and 1-2 shorter breaks of 15 min each. Calls are handled consecutively with little time in between; on average 4.3 (SD = 4.1) min between calls, including time for case documentation but excluding breaks >15 min. Most calls are short, median 6.3 (IQR = 5.3) min.

Taken together, we see this as an ideal setting for observing behavioral correlates of fatigue, because the work is hard and repetitive but still requires qualified assessment.

**Identifying best comparison points for high vs. low fatigue.** In line with the previous literature, we proxy for fatigue using an observational correlate of cumulative mental effort. This means that we identify instances where nurses are plausibly tired (depleted) due to having worked hard for a prolonged period of time, e.g., at the end of their shift; and conversely, instances where nurses are plausibly less depleted, e.g., because their shift just started or they just had a lunch break. Cumulative mental effort is a key component in the conceptual basis of decision fatigue (see Introduction). As an alternative, we could have collected actual measures of subjective fatigue (e.g., prompted self-reports), but this would not have been possible without compromising on key features of our study design, most importantly sample size, and we would have been worried about experimenter demand effects.

There are two suitable comparison points for fatigue in our data/ setting. The first is a before vs. after breaks comparison because on the one hand breaks are distributed over the work day, but at the same time each comparison is confined to decisions taken during a shorter period of time (Fig. 2). Confounding due to time of day variation in calls is thus unlikely in this case. The second suitable comparison point emerges due to the specific scheduling procedure (staggered, rolling), which produces a quasi-experimental setting in the early afternoon (Fig. 1). Inside this time window, workers on the morning shift are on the back of almost a full workday (high fatigue) whereas workers on the afternoon shift just started (low fatigue), but still, their calls are drawn in a pseudorandom order from the same pool of patients. Figure 3 shows the substantial separation in cumulative workload between morning and afternoon workers that took calls inside the same time window. Morning workers had on average spent 6.5 h (SD = 0.8, *n* = 4948 calls) at work and handled 25.6 calls (SD = 7.8, *n* = 4948 calls) compared to 0.6 h (SD = 0.5, *n* = 4788 calls) and 4.2 calls (SD = 3.2, *n* = 4788 calls) for afternoon workers (data source: target data). Corresponding statistics in the pilot data: Morning workers had on average spent 6.5 h (SD = 0.8, *n* = 6221 calls) at work and handled 26.9 calls (SD = 9.1, *n* = 6221 calls) compared to 0.6 hours (SD = 0.6, *n* = 6451 calls) and 4.6 calls (SD = 3.6, *n* = 6451 calls) for afternoon workers. Thus, if there are relevant effects from fatigue on decision-making and judgment, as the concept decision fatigue presupposes, we should arguably see them here.

In summary, we tested for differences in outcomes following our hypotheses (see "Introduction") in two different subsamples (comparison points):

a. Overlap AM-PM shift ca 13:00–17:00. Quasi-experiment.
b. Intense call sequences before vs. after breaks (8–40 min)

Below we give more details on how these subsamples were selected.

### Sampling plan

**Time period and data access.** We collected data from the same healthcare providers as in the pilot data for the period October–December 2021 & 2022, excluding Christmas (exact dates: Sept 27–Dec 19, 2021 and Sept 26–Dec 18, 2022). The pilot data covered the period Oct 21–Dec 22, 2019, and Oct 19–Dec 20, 2020. We did not have access to the new data before the Stage 1 protocol was accepted in principle and date stamped. A digital infrastructure provider (DIP) hosts the registry databases and permission is needed for access and use of data for research or other purposes. This DIP thus acts as a credible data gatekeeper.

**Data inclusion and exclusion.** Based on the pilot data sample size, we expected a stage 2 sample size of at least *n* = 270,000 new phone calls. The expected sample size was computed using a simple linear extrapolation from the pilot data; the time period would increase with 33% (from 18 weeks to 24 weeks) and therefore we expected the sample size to also

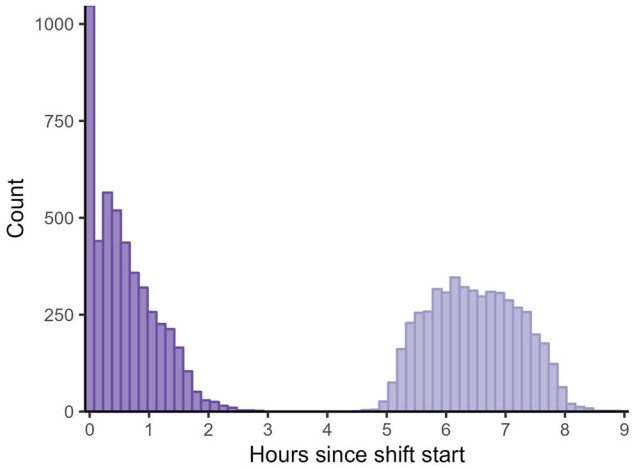
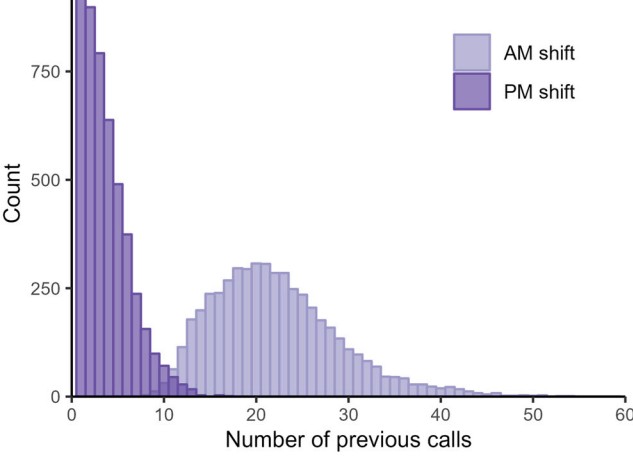

**Fig. 3 | Separation in cumulative workload for observations included in the AM-PM shift overlap subsample.** *Left*: Distribution of time spent at work by type of shift (AM vs. PM) for calls handled during the overlap time window. [*n* = 9736] *Right*: Distribution of number of previous calls on the same day by type of shift (AM vs. PM) for calls handled during the overlap time window. AM shift indicated in light purple, PM shift in purple. [*n* = 9736]. See "Data inclusion and exclusion" subsection for details on how the AM-PM shift overlap subsample was selected.

increase with around 33%, i.e., from *n* = 203,100 to approximately *n* = 270,000. The achieved sample size was smaller than expected, but still substantial, at *n* = 231,076, *k* = 174. This sample still met the pre-registered power requirements on three of four planned main tests, and it was slightly below criterion on the fourth test (see the Sensitivity power analysis subsection below for details). The main reason why the achieved sample size was smaller than expected is likely that a substantial part of the pilot data coincided with one of the most busiest periods during the COVID-19 pandemic (autumn 2020), which we did not take into account when setting expected sample size; in retrospect, we should probably have anticipated a less-than-proportional increase in sample size vis-à-vis time period covered. All phone calls from the stage 2 sample period were included in the main sample and we then systematically excluded data in order to reach each analysis subsample (overlap and breaks, respectively). The pilot data was not included in stage 2 description or analysis.

There is no designated shift or workday information in the raw data, i.e., there are no variables purposefully showing shift start, end, or break times. We therefore define shifts, breaks, and other needed characteristics based on observed sequences of phone calls (for which we have date:hh:mm:ss data) handled by each nurse on each day. This is a suitable approach, but also note a limitation for the analysis of calls around breaks, since non-call tasks, which would still constitute work, e.g., paperwork, training, could be mistaken for breaks.

We defined exact rules for data exclusion in advance (based on the pilot data) for the creation of each analysis subsample. Some of them may seem unnecessary crude or arbitrary but we took this approach to ensure reproducibility and to not overfit on the pilot data. We describe general considerations for creating each subsample here; the detailed rules can be found in the associated online data handling description/code documents available on the project's OSF repository. These rules were developed using the pilot data and we followed them completely at stage 2.

a) Overlap AM-PM shift. Here we want as much separation as possible in time spent at work (long for the AM shift, short for the PM shift) but also as large a sample as possible. For example, choosing only those nurses who start very early would mean increased separation (longer time until the overlap period) but at the cost of getting a smaller sample, because those who start later but are still in the AM shift would be excluded. We used the following main rules: Consider only decisions taken between 13:00–17:00 on Mon–Fri for workers who either take their first call 6:00–9:00 and their last call after 13:00 (AM worker), or take their first call 13:00–18:00 and their last call after 19:00 (PM worker). Then also exclude workers who take <12 calls because they are likely working on other tasks (e.g., admin) that we do

not have data on. Further, we divide the relevant time window (13:00–17:00) in 30 min time bins (thus eight in total) and keep only those bins with >100 observations for each category of workers (AM and PM shifts, respectively). Applied to the pilot data these rules resulted in a final sample size of *n* = 9730 calls to *k* = 166 nurses; applied to the target data (stage 2), using these rules resulted in a final sample size of *n* = 9648 calls to *k* = 143 nurses.

b) Intense call sequences before vs. after breaks. Here we want sufficiently long and intense sequences, meaning many calls with little time in-between for rest, but also as large a sample as possible. We used the following main rules: Consider only sequences where a nurse handled at least five calls before and five calls after a break 8–40 min, and there are at most five min between each call. The exact values used here are arbitrary (unless changed substantially), e.g., using 7 or 9 min instead of 8 min as lower cutoff for a break should not make a difference; the important part for us was to decide on and specify these values *exactly and in advance*, in order to credibly restrict analytic flexibility later on when analyzing the new data. These rules resulted in a final sample size of *n* = 5120 calls to k = 116 nurses in the pilot data; and *n* = 4940 calls to *k* = 77 nurses in the target (stage 2) data.

**Power analysis.** Determining target sample size for stage 2 data, our objectives were to ensure (i) sufficient probability to detect a relevant effect, assuming it exists, and low probability of generating misleading evidence, and (ii) similarity with the pilot data in terms of work environment, scheduling and call composition to ensure that predefined criteria for data inclusion and exclusion and analysis will be as applicable and relevant as possible. Based on these considerations, we planned to collect data from the same healthcare providers as for the pilot data for the period October–December 2021 & 2022 (excluding Christmas; see above for exact dates). We expected larger analysis samples than in the pilot data.

Relevant effect size can be determined by combining information from previous estimates in Persson et al.[13], where we have the data and know the setting, with our knowledge about the current setting and with relevant decision-making patterns in the pilot data. Persson et al.[13] estimated that the proportion of patients scheduled for operation at an orthopedic clinic declined from approximately .4 to .2 from first to last patient during a half-day shift. Assuming that effects are overestimated in the previous literature, we aimed for good power to obtain compelling evidence for an effect of half the magnitude in Persson et al., which in generic terms is equivalent to a Cohen's *d* = 0.20, or odds ratio = 1.5 (0.10 proportion difference at a base rate of 0.40).

We simulated power for our design using a *fixed-n* approach for Bayes Factor analysis[65] and taking into account that total sample size $n$ is a function of both the number of nurses $k$ (clusters) and the number of calls to each nurse $m$ (cluster size), and that observations within a cluster may be correlated[66]. Assuming a stage 2 sample of $k > 150$ nurses handling $m > 10$ calls each in both *high* and *low* fatigue conditions, we approximated a paired design for an effect of fatigue using a one-sample test for mean difference >0 at *effective statistical sample size* $n_e = 1000$. This sample size was computed using $k \times m \times (DE)^{-1}$, where the design effect $DE$ equals $1 + (m - 1) \times ICC$ and intra-cluster correlation $ICC$ was set to 0.05[66].

With this setup, we generated 10,000 samples for an assumed positive effect $\sim N(0.2, 1)$ and for each sample computed relative evidence for H+: $\delta > 0$ vs. a point null H$_0$: $\delta = 0$ using one-sided Bayes Factor and analysis prior Cauchy with $r$ scale $\sqrt{2}/2$. Using BF > 10 as criterion for conclusive evidence, 99.9% of these samples were correctly classified as providing evidence in favor of H+ and the false negative rate was 0%. Similarly, we generated another 10,000 samples for an assumed null effect $\sim N(0, 1)$ and again computed a one-sided Bayes Factor for H+: $\delta > 0$ vs. a point null H$_0$: $\delta = 0$ using the same analysis prior as before. This resulted in correct classification (relative support for H$_0$) in 83.9% of samples (<1% misclassified) using BF > 10, and 92.7% correct at BF > 5. Taken together, these computations suggested that our design had a high probability of generating strong evidence that would move the field forward.

*Sensitivity power analysis.* Even though we followed our sampling plan completely, the achieved sample was smaller than expected. We therefore conducted a sensitivity power analysis for the targeted effect size at the achieved sample. We did this for our four main tests, i.e., for both *Urgency* and *Default* each in the Overlap and Breaks analysis subsamples. We took target effect size as given but used $k$, $m$ (sample median), and $ICC$ from the respective achieved samples.

*Overlap sample.* For *Urgency*, we have $k = 143$, $m = 24$, and $ICC = 0.056$. For an assumed positive effect (same as above), using BF > 10 as criterion for conclusive evidence, 100% of simulated samples were correctly classified in favor and the false negative rate was 0% (0% inconclusive). For an assumed null effect, 87.7% were correctly classified and 0.1% false positives (12.3% inconclusive). For *Default*, we have $k = 135$, $m = 14$, and $ICC = 0.003$. For an assumed positive effect, 100% of simulated samples were correctly classified in favor and the false negative rate was 0% (0% inconclusive). For an assumed null effect, 88.8% were correctly classified and 0.1% false positives (11.2% inconclusive).

*Breaks sample.* For *Urgency*, we have $k = 75$, $m = 10$, and $ICC = 0.063$. For an assumed positive effect, 91.2% of simulated samples were correctly classified in favor and the false negative rate was 0% (8.8% inconclusive). For an assumed null effect, 76.2% were correctly classified and 0.1% false positives (23.7% inconclusive). For *Default*, we have $k = 73$, $m = 7$, and $ICC = 0.023$. For an assumed positive effect, 88.2% of simulated samples were correctly classified in favor and the false negative rate was 0% (11.8% inconclusive). For an assumed null effect, 74.8% were correctly classified and 0.2% false positives (25.0% inconclusive).

In summary, despite the smaller sample, the power requirements were still satisfied for Urgency in both the Overlap sample and in the Breaks sample, and for Default, it was satisfied in the Overlap sample but slightly below criterion in the Breaks sample.

### Analysis plan

**Dependent variables.** We define two dependent variables (DVs) corresponding to our hypotheses (see Introduction). *Default* is a binary indicator (1 = yes, 0 = no) for assigning the modal level of urgency to a call, where modal is defined as the level of urgency most often chosen by that particular nurse for that particular type of patient. Type here refers to call category (predefined categories, e.g., chest pain) and the mode is calculated for all such calls to each nurse excluding the call currently under consideration and provided there are at least 8 such calls. Observations with multiple modes are dropped from analysis. *Urgency* is a

numeric variable for assigned level of urgency, range 0–4, where higher values indicate greater urgency to get in-person medical attention (0 = "wait", 1 = "in a couple of days", 2 = "within 24 h", 3 = "urgently", 4 = "immediately").

**Independent variables.** We define two independent variables that identify observations as plausibly high or low fatigue. Our main analysis approach is based on statistical comparisons of outcomes across levels of these variables. *Before break* is a binary indicator for a call occurring before (=1) or after (=0) a break, provided the call meets our criteria for inclusion in a before/after-break sequence (see Data inclusion and exclusion for description of these criteria). This variable is relevant for analyses in the breaks subsample. We assume fatigue is higher before than after a break. *Morning* is a binary indicator for a call to a nurse working the morning (AM) shift (=1) or to a nurse working the afternoon (PM) shift (=0), provided all criteria are met for inclusion in the AM-PM overlap subsample (see Data inclusion and exclusion for description). This variable is relevant for analyses in the overlap subsample. We assume fatigue is higher for nurses on the morning shift than on the afternoon shift. In addition, we define *time indicator variables* for inclusion (=1) in relevant 30 min time bins inside the overlap time window. For example, *Time 1* indicates whether or not a particular call occurred between 13:00 and 13:30 (in the overlap sample), and so on. We include these as independent control variables to account for potential variations over time inside the overlap time window.

**Planned data description.** We first describe some general patterns in the data, including overall urgency ratings. We then show unadjusted results for our main DVs in each analysis subsample and comment on them in the text. For descriptive purposes, we also show unadjusted urgency ratings by time of day. Display items are included in the main text or in Supplementary Material depending on space constraints.

**Models for main confirmatory tests.** a) Overlap AM-PM shift. For *Default*, we fit a Bayesian generalized mixed effects model for binary dependent variables on *Morning*, with random intercepts for nurses and time indicator variables (=1 for inclusion in relevant 30 min bin) as fixed factors. For *Urgency*, we fit a Bayesian generalized mixed effects model for continuous dependent variables on *Morning*, with random intercepts for nurses and time indicator variables (=1 for inclusion in relevant 30 min bin) as fixed factors.

b) Intense call sequences before vs. after breaks. We use a similar analysis approach as in a): For *Default*, we fit a Bayesian generalized mixed effects model for binary dependent variables on *Before break*, with random intercepts for nurses. For *Urgency*, we fit a Bayesian generalized mixed effects model for continuous dependent variables on *Before break*, with random intercepts for nurses.

We use weakly informative analysis priors, with a Normal distribution with mean zero and standard deviation = 10 for both the fixed effects and the intercept, and a Cauchy prior with $r$ scale $\sqrt{2}/2$ for the standard deviation of random effects.

Analysis codes are available on the project's OSF repository.

Assumptions underlying these models were not formally tested.

**Evaluation.** Interpretation and overall assessment are based on one-sided Bayes Factor (BF) hypothesis testing for effects in the predicted direction (H$_+$) vs. a point null hypothesis (H$_0$) of no effect. Subsequently, we also interpret 95% Credible Intervals for the relevant estimated parameters (Morning and Before break) transformed into a relevant measure for effect size. For Default, we use odds ratio as effect size and thus take the exponentials of the CrI endpoints for the model coefficient estimates, and for Urgency, we use Cohen's $d$ and thus divide the CrI endpoints with the sample standard deviation of the DV.

In total, we have four main models to evaluate (as described above; two hypotheses about decision fatigue each tested in two different subsamples,

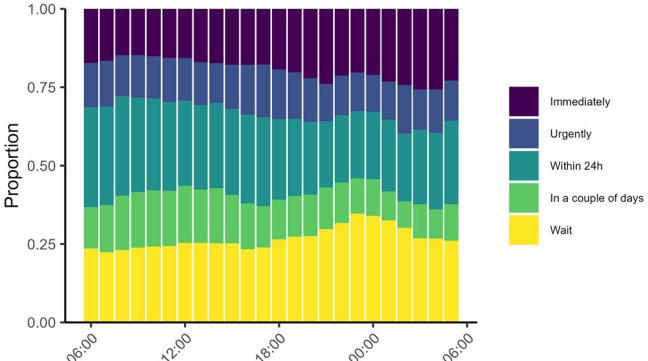

**Fig. 4 | Time of day and assigned urgency.** Each bar shows the distribution of urgency ratings for calls during that hour. Higher urgency ratings later in the day would be expected under decision fatigue. This analysis is not part of our main test battery [$n = 125{,}587$].

overlap and breaks). For each model, we compute a two-sided $BF_{10}$ for the full model vs. the restricted model that excludes the relevant independent variable. We then approximate a one-sided $BF_{+0}$ (our main basis for evaluation) as $BF_{+0} = 2 \times (1 - p_{>}) \times BF_{10}$, where $p_{>}$ stands for the one-sided p-value, following Morey and Wagenmakers[67]. We obtain the one-sided p-value for each model using the estimated coefficient and standard error in a conventional large-sample approximation.

Interpretation of results from each test (each model). Each test is evaluated on the basis of a one-sided Bayes Factor as described above. We interpret $BF_{+0}$ as the relative strength of evidence for $H_+$ (directional hypothesis for decision fatigue) vs. $H_0$ provided by the data. For example, $BF_{+0} = 5$ would indicate that the observed data are five times more likely under $H_+$ than $H_0$.

Assessing support for each hypothesis. Each hypothesis is tested in two different subsamples, thus two Bayesian hypothesis tests as described above. We assess support for each hypothesis based on a classification of the two associated one-sided BFs using BF > 10 as a primary criterion for conclusive evidence: The hypothesis is *supported* if $BF_{+0} > 10$ for at least one (of the two) associated tests; the hypothesis is *not supported* if instead the results favor $H_0$ over $H_+$, for which we required that both BFs satisfy $BF_{+0} < 1/5$ AND at least one of them $BF_{+0} < 1/10$; and otherwise we count evidence for the hypothesis as inconclusive.

Overall conclusion. Final assessment of support for decision fatigue is based on the aggregated results from the four underlying Bayesian hypothesis tests as follows: *Support* for decision fatigue if at least one hypothesis is supported; *no support* for decision fatigue if both hypotheses are classified as "not supported"; otherwise *inconclusive*.

Preregistered exploratory questions. We planned room for interpreting the strength of evidence for the arrived-at conclusion, based on magnitude of the (one-sided) BFs, and direction and magnitude of the relevant estimated parameters indicated by the 95% credible intervals from the models. If our overall conclusion was 'inconclusive' (per our classification above), we still planned to interpret these measures directly (BFs and range for estimated parameters) to increase the precision of our conclusion, if possible.

### Reporting summary
Further information on research design is available in the Nature Portfolio Reporting Summary linked to this article.

## Results
There were 160,357 incoming patient calls during the study period. Out of these, 143,970 resulted in an assigned urgency rating, with 16.1% deemed highly urgent and thus in need of immediate medical attention, 32.1% were

assigned the lowest urgency rating (wait); and the remaining 51.7% were assigned one of the three intermediate urgency ratings. 18.0% of incoming calls resulted in consultancy. Mean call duration was 7.6 min (SD = 5.8). Calls during evenings and nights (19:00–06:00) were slightly longer and resulted in higher urgency ratings (mean ± SE call duration, 8.2 ± 0.027 min vs. 7.4 ± 0.015 min; highest urgency, 20.6 ± 0.20 vs. 14.4 ± 0.11). There was substantial variation among individual nurses in assigned urgency ratings. For nurses with >200 calls during the study period (98.9% of relevant incoming calls), prop. highest urgency ranged from 11.5% to 19.5% (1st to 3rd quartile). The proportion of calls resulting in external consultation ranged from 13.8% to 22.7%. This likely reflects both individual differences in work schedules (e.g., some nurses work mostly nights) and in their approach to cases.

### Descriptive results for decision fatigue
A general approach used in the previous literature has been to use time of day as a proxy for fatigue. Here we show results for assigned urgency at different times of day (Fig. 4) for descriptive purposes but we do not assign much evidential value to these results because they are likely confounded by temporal patterns in call composition (see Supplementary Note 1 for more discussion about this). The figure shows that higher urgency ratings appear to be chosen more often later in the day. The proportion of calls assigned the highest urgency rating (immediate) was 14.8% in the morning (around 08:00) and it increased to 19.3% and 21.1% respectively for calls in the evening and at night (around 18:00 and 00:00, respectively). A similar pattern emerged for calls taken later during a work shift, as indexed by call order; during afternoon and night shifts (but not morning shifts), the likelihood of assigning highest urgency rating seemed to increase with call order (Supplementary Fig. 3).

In the *overlap* subsample, all incoming calls are plausible as-if randomly assigned a nurse with either a high or low level of fatigue. Descriptive results for this subsample are shown in Fig. 5. Both panels show the distributions of the relevant dependent variable (*Urgency* and *Default*, respectively) averaged by nurse, for calls inside the overlap time window. We can see that both variables appear to be approximately similarly distributed across morning (AM) and afternoon (PM) shifts, except a somewhat increased concentration around the sample means for workers in the morning shift.

In the *breaks* subsample, we proxy for fatigue comparing decisions at the end of an intense call sequence, approaching an 8–40 min break, with decisions after the break. Descriptive results for this subsample are shown in Fig. 6. The left panel shows the distribution of assigned urgency ratings by relative distance (number of calls) to the break and the right panel shows the overall proportion of calls where the assigned urgency rating was in line with that worker's default (for that particular call type), also shown by relative distance to the break. We had hypothesized higher urgency ratings before compared to after the break, and increased use of personal defaults, but we see no clear such patterns. The distributions for both variables look approximately similar across the different call orders and before compared to after the break. Overall, we see no clear patterns in these data that would be indicative of decision fatigue.

### Main tests for decision fatigue
We test statistically for effects consistent with decision fatigue for each DV in both subsamples using a Bayesian generalized mixed effects model for binary or continuous dependent variables (*Default* and *Urgency*, respectively), with random intercepts for nurses. The results are shown in Tables 2–3. We had hypothesized that workers on the morning shift would choose more in line with personal defaults and assign higher urgency ratings on average, compared to workers on the afternoon shift, during the overlap time window. However, this is not what we see in the data; the point estimates on *Morning* are small for both *Default* and *Urgency* (odds ratio = 1.07 and Cohen's $d = 0.05$, respectively), and the posterior distributions generally have little mass on effects of relevant magnitude in the predicted direction. Similarly, we hypothesized that workers approaching a break

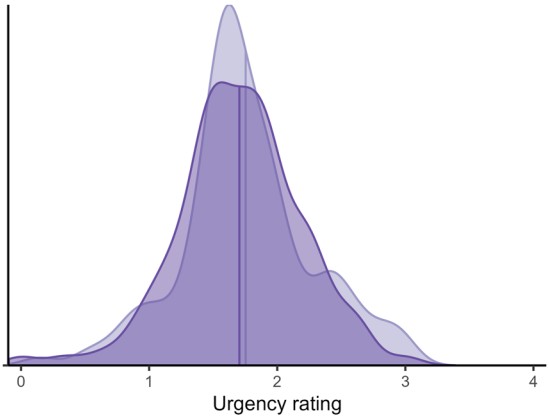
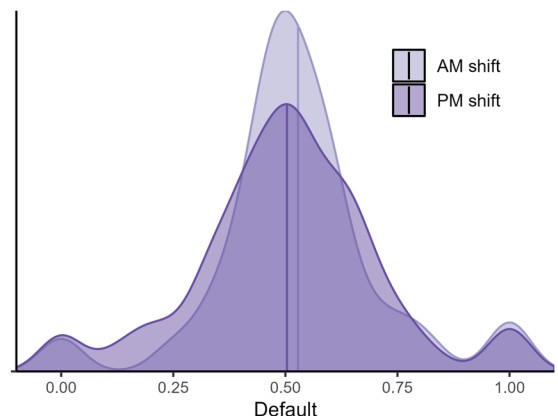

**Fig. 5 | Outcomes for morning vs. afternoon shift in the overlap subsample.** *Left:* Data and distribution for *Urgency*, showing the average (for each nurse) urgency rating assigned for calls during the overlap time window, separated by AM and PM shifts. We hypothesized higher urgency ratings on average for workers on the AM shift. [*n* = 9648]. *Right:* Data and distribution for *Default*, showing the proportion of calls a particular nurse chose the most common urgency rating (personal modal) for that type of call during the overlap time window, separated by AM and PM shift. We hypothesized higher proportion *Default* for workers on the AM shift. Vertical lines denote mean values. AM shift indicated in light purple, PM shift in purple [*n* = 6158].

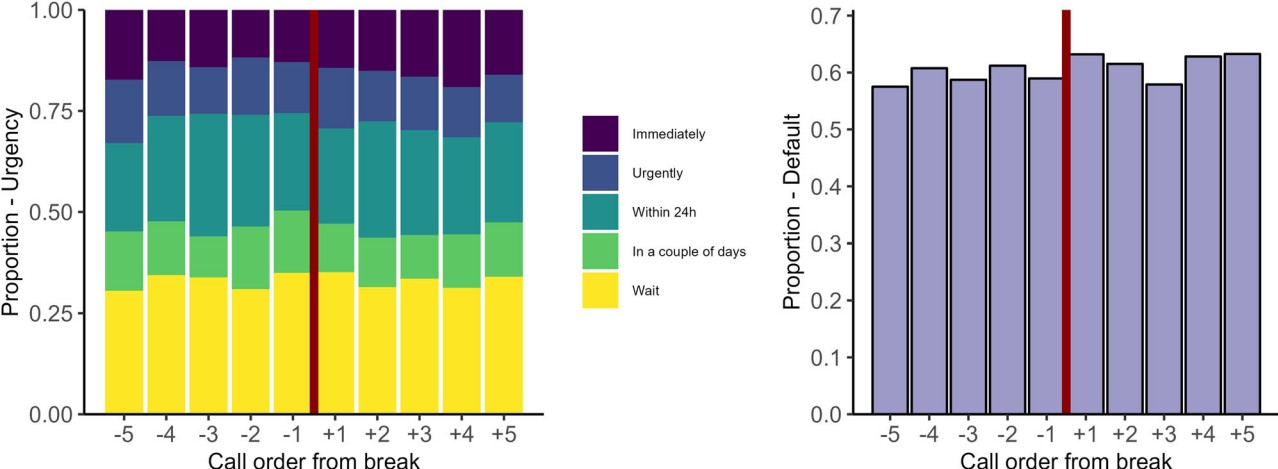

**Fig. 6 | Outcomes before vs. after breaks in the breaks subsample.** *Left:* Assigned urgency at different call orders relative to the occurrence of an 8–40 min break. Purple means highest urgency (immediately), yellow lowest urgency (wait). We hypothesized higher urgency ratings before compared to after breaks. Calls were chosen on the basis of max 5 min between each call (except break) for each call sequence. [*n* = 3794]. *Right:* Overall proportion default at different call orders relative to the occurrence of an 8–40 min break. We hypothesized increased use of personal defaults before compared to after breaks [*n* = 2779].

would choose more in line with decision fatigue than right after a break, but estimated effects here go in the wrong direction (e.g., 9% decrease in the odds of choosing in line with personal defaults), and posterior distributions again have little mass on effects of relevant magnitude in the predicted direction.

The associated two-sided Bayes Factors for the shown models vs. a null model where the relevant independent factor was excluded indicate more support for $H_0$ over $H_1$ in both samples, with $BF_{01} > 43$ for both *Urgency* and *Default* in the overlap sample, and >6 in the breaks sample.

## Evaluation
Results from our main tests organized by hypothesis are summarized in Table 4. We had hypothesized that the likelihood to choose in line with personal defaults and assigned urgency ratings should increase with fatigue, and thus show up as positive one-sided Bayes Factors $BF_{+0} > 1$ when computed from our main models. We interpret $BF_{+0}$ as the relative strength of evidence for $H_+$ (the directional hypothesis for decision fatigue) vs. $H_0$ provided by the data. Looking at the table we can see that our results in fact provide strong support for the null, with $BF_{+0} \leq 0.045$ indicating that in each case, the observed data are >22 times more likely under $H_0$ than $H_+$.

Thus, none of the hypotheses are supported, and overall we find no evidence for decision fatigue.

## Preregistered exploratory questions
Here we show results for call duration and proportion of calls resulting in external consultation. These variables are interesting since effects could plausibly be rationalized as decision fatigue ex-post, but at the same time, it is not obvious what direction of effects to expect ex-ante. We therefore decided beforehand to treat these analyses as exploratory. We use the same analysis approach as for our main outcome variables. Statistical results are presented in Supplementary Table 1.

The mean estimate for effect on proportion consult was close to zero in both samples and thus on average seemed little influenced by level of fatigue (overlap, odds ratio = 0.97, 95% CrI, 0.85, 1.11; breaks, odds ratio = 1.05, 95% CrI, 0.91, 1.21). For call duration, there were small effects but in opposite directions in the two samples, with calls on average slightly longer going into a break compared to right after (Cohen's *d* = 0.05, 95% CrI, 0.01, 0.09); and calls on average slightly shorter for morning workers compared to afternoon workers during the overlap time window (Cohen's *d* = −0.05, 95% CrI, −0.09, −0.01). However, none of these estimates are consistent

**Table 2 | Results from main tests for decision fatigue in the overlap sample**

|  | [1] Default Coef. | Odds ratio | [2] Urgency Coef. | Cohen's $d$ |
|---|---|---|---|---|
| *Morning* | 0.063 (0.062) | 1.07 [0.94, 1.20] | 0.071 (0.038) | 0.050 [−0.002, 0.102] |
| Time 3 | 0.056 (0.084) |  | 0.040 (0.047) |  |
| Time 4 | 0.174 (0.087) |  | 0.094 (0.049) |  |
| Time 5 | 0.124 (0.092) |  | 0.139 (0.052) |  |
| Time 6 | 0.115 (0.094) |  | 0.178 (0.055) |  |
| Intercept | -0.079 (0.085) |  | 1.596 (0.056) |  |
| N observations | 6158 |  | 9648 |  |
| N clusters | 135 |  | 143 |  |
| Bayes Factor, BF$_{10}$ | 0.009 |  | 0.023 |  |

Table notes: In column [1], for *Default*, we used a Bayesian generalized mixed effects model for binary dependent variables, with random intercepts for nurses and time indicator variables (*Time 2* was reference category) as fixed factors, and we used weakly informative priors. Standard errors are shown in parentheses. In column [2], for *Urgency*, our approach was the same except here we used a Bayesian generalized mixed effects model for continuous dependent variables. The sample standard deviation for *Urgency* was 1.42, used in the calculation of Cohen's $d$. Odds ratio and Cohen's $d$ are given as converted point estimates and associated 95% credible interval. We hypothesized a positive effect (and non-negligible magnitude) of *Morning* in both models. BF$_{10}$ is the two-sided Bayes Factor for the full model vs. the restricted model that excludes *Morning*.

**Table 3 | Results from main tests for decision fatigue in the breaks sample**

|  | [1] Default Coef. | Odds ratio | [2] Urgency Coef. | Cohen's $d$ |
|---|---|---|---|---|
| *Before break* | −0.097 (0.082) | 0.91 [0.77, 1.06] | −0.059 (0.045) | −0.041 [−0.103, 0.020] |
| Intercept | 0.566 (0.085) |  | 1.719 (0.065) |  |
| N observations | 2779 |  | 3794 |  |
| N clusters | 73 |  | 75 |  |
| Bayes Factor, BF$_{10}$ | 0.163 |  | 0.11 |  |

Table notes: In column [1], for *Default*, we used a Bayesian generalized mixed effects model for binary dependent variables, with random intercepts for nurses, and we used weakly informative priors. Standard errors are shown in parentheses. In column [2], for *Urgency*, our approach was the same except here we used a Bayesian generalized mixed effects model for continuous dependent variables. The sample standard deviation for *Urgency* was 1.43, used in the calculation of Cohen's $d$. Odds ratio and Cohen's $d$ are given as converted point estimates and associated 95% credible intervals. We hypothesized a positive effect (and non-negligible magnitude) of *Before break* in both models. BF$_{10}$ is the two-sided Bayes Factor for the full model vs. the restricted model that excludes *Before break*.

**Table 4 | Main assessment of decision fatigue**

|  | Tested in sample | BF$_{+0}$ Relative evidence for $H_+ : \delta > 0$ vs. $H_0 : \delta = 0$. |
|---|---|---|
| Hypothesis 1 Default more likely under fatigue. | Overlap | 0.015 |
|  | Breaks | 0.038 |
| Hypothesis 2 Urgency ratings higher under fatigue. | Overlap | 0.045 |
|  | Breaks | 0.021 |

Table notes: The one-sided Bayes Factor BF$_{+0}$ quantifies relative evidence for the directional hypothesis following decision fatigue (e.g., default more likely under fatigue) vs. the null hypothesis of no difference in outcome depending on fatigue. BF$_{+0}$ was approximated using the two-sided BF$_{10}$ (and associated one-sided $p$-value) computed for our main models (reported in Tables 2–3) using the approach in Morey and Wagenmakers[57].

with effects of substantial magnitudes. Overall, these additional results do not speak in favor of decision fatigue.

## Discussion

We conducted a strong, focused empirical test of decision fatigue as a practically relevant concept for understanding repeated or effortful decision-making. We used real decisions from healthcare and reasoned that decision fatigue should show up i) in decisions among workers nearing the end of a work shift, compared to workers who had just begun work, and ii) in decisions made before vs. after a break. In both cases, we hypothesized increased use of heuristics, in particular, that choices would converge toward personal defaults ("typical judgment" for this type of patient), and higher assigned urgency ratings. We used a quasi-experimental, "overlapping time

window" approach to adjust for patient assignment and time of day in our main analyses. The context was a national telephone triage and medical advice service in Sweden ($n = 231,076$ phone calls involving $k = 174$ specialized nurses in the target data). We had previously collected a large pilot data set to understand the work environment and develop and refine hypotheses, and eventually pin down a suitable analysis pipeline for the target data (Stage 1 protocol).

Using this approach, we found no support for decision fatigue. Bayesian generalized mixed effects models consistently indicated relative support for the statistical null hypothesis of no difference in outcome depending on fatigue, with one-sided Bayes Factors >22 in favor of the null for all four main confirmatory tests. Point estimates for effects were generally small and the 95% credible intervals had little mass on estimates of sufficient magnitude in the expected direction. Results from the preregistered exploratory analyses were similar, with no clear indication of decision fatigue.

Decision fatigue as a concept builds on the reasonable idea that sooner or later when people get tired or even depleted, there will be some kind of downstream effect on behavior. However, the concept also seems to suppose that this effect should set in surprisingly fast and certainly with relevance inside a couple of hours of hard work or over the extent of a full working day. Our results suggest that this may not be the case and that empirical patterns that look like decision fatigue may instead be due to other factors. From a practical perspective, these findings are reassuring. An implication for policymaking is that generic concerns about decision fatigue should have little influence on policy in healthcare or elsewhere, without further context or organization-specific qualification. There may of course be other, well-founded grounds for mitigating potential fatigue in workers; our argument here is with the supposed behavioral implications of fatigue, not with the experience of fatigue itself and the potentially adverse consequences this may have for workers. Also, policy

may still have a role to play in addressing other, potentially related behavioral factors such as boredom or lack of motivation.

From a theoretical perspective, our results are a little disappointing. Decision fatigue has an intuitive feel to it and is likely something many people can relate to from experiences in their daily lives, but our findings quite convincingly speak against decision fatigue as a strong domain-general effect. One can of course speculate whether the relationship between fatigue and information processing or cognitive effort avoidance could be more subtle or complicated than previously thought. It might be the case that there is a real effect that is more nuanced, weaker, potentially heterogeneous or context-dependent, but to some extent, these things can always be said in defense of a theory, and they are hard to refute empirically. What can be said here is that we looked long and hard for decision fatigue but couldn't find it.

There is also a methodological point to be made about how easy it can be to find an effect that may look like decision fatigue, or more generally find any type of effect that presupposes a sequential data pattern. A case in point here is that our descriptive analyses in both the pilot and target data showed what seemed like a non-negligible time-of-day effect on behavior. Higher urgency ratings were assigned more often later in the day. It would have been easy to build a case (in good faith) where this pattern was taken as prima facie evidence of decision fatigue, pointing to real and practically relevant effects. However, that interpretation would likely be wrong because the observed association between time of day and assigned urgency is likely confounded by other time-of-day dependent effects, for example, composition of call types or organizational factors like opening hours of primary care facilities. For these reasons, care should be taken if using time of day or call order as proxies for fatigue when the goal is to estimate influence on decisions. Both variables may induce confounding, but to what extent of course will depend on setting and overall analytic strategy. In our case, call order is problematic because it is correlated with time of day. Maybe this problem could have been alleviated by exploiting the staggered pattern of shift start times, but possibly not well enough, also considering there were other more suitable proxies for fatigue available in the data.

Our approach had several strengths and benefitted from a near-ideal decision environment for testing decision fatigue. Work in this setting is both hard and repetitive, yet qualified, and the variation in scheduling means that level of fatigue could be regarded as near random for some segments of the data. This also helped account for some classical confounding factors discussed in the fatigue literature, including boredom, sleepiness, and lack of motivation. The worry would be that these factors might lead to disengagement and thereby mask any real effects coming from decision fatigue[68]. However, with our design such confound is unlikely, because workers at low-fatigue comparison points were performing the same type of qualified work, just at lower cumulative intensity; and external factors like type of calls were approximately similar. Sleepiness is similarly accounted for, since comparisons were always close in time, although some caution might be warranted for analysis of the breaks subsample during night, where it is possible that sleepiness is more pronounced going into a break than right after.

Along similar lines, high motivation could also lead to confounding. The worry would be that motivation, when high and under the right circumstances, may help people overcome fatigue even after bouts of cognitive activity[69]. Tasks that are inherently fun or excessively rewarding, and where people have full situational control over how much effort to exert, are thus according to this view unlikely to induce fatigue on a substantial level[70]. However, at work people often do not have full situational control to adjust their effort seamlessly; there are expectations to perform to a certain level, and if demand is high, hard work is likely needed. From this perspective, using archival/field data from settings where work is demanding, repetitive, and requiring qualified assessments on back-to-back cases, like in our case, seems suitable for testing decision fatigue.

## Limitations

There are some limitations that should be kept in mind when interpreting our results. First, our hypotheses and the underlying behavioral theory represent one, but not the only, way to conceptualize decision fatigue. In our view, it is a coherent and the most sensible approach for our setting, although the description (and application of rational inattention theory) used here arguably overemphasizes the conscious rationality of the effect. Second, our data are based on work in a specific context inside one specific organization. Although there is a good case for assuming a certain degree of domain generality when it comes to work tasks (with respect to decision fatigue), other organization-specific factors like training or guidelines might play a role. Also, nurses have access to a symptom-based, scaffolding decision support system. Using this system is part of the work process when handling calls. It would have been interesting to analyse variations in how this system is used both within and between nurses, but we did not have access to this data. Generalization of our findings should be made keeping these limitations in mind. On a more detailed level, and as discussed in the Methods section, two other relevant caveats are that we did not attempt to collect actual measures of fatigue (e.g., self-reports) and that some central case characteristics, like breaks and work shifts, had to be inferred by mapping observed data patterns to aggregate information about workdays.

## Conclusion

We found no evidence for decision fatigue using a large sample of real decisions from healthcare. Whereas these findings don't preclude the existence of a weaker version of decision fatigue or more context-specific effects, they cast serious doubt on the empirical relevance of decision fatigue as a domain general effect for sequential decisions in healthcare and elsewhere.

## Data availability

All data needed to reproduce the study's main analyses (main tests and evaluation) is publicly available on the project's OSF repository at https://osf.io/8tvfs/ [https://doi.org/10.17605/osf.io/8tvfs]. These data contain all variables and all observations (at full disaggregation but without participant identifiers) used in the main analyses for each of the two analysis subsets, overlap and breaks respectively. For ethical reasons (participant integrity), data and features not used in the planned final analyses are deleted before uploading, and there is no way for readers to access the full raw data. Data from the pilot study (same structure) is available on the same OSF repository (stage 1 folder).

## Code availability

All analysis codes needed to reproduce the study's main analyses as well as descriptions and codes used for creating each analysis subsample ("data handling") are publicly available on the project's OSF repository at https://osf.io/8tvfs/ [doi: 10.17605/osf.io/8tvfs]. The corresponding files for creating and analysing the pilot data are available via the same link (stage 1 folder), except the data handling descriptions for the pilot data do not contain codes. These files, for pilot-data main analysis and data handling, were uploaded before the stage 1 protocol was accepted in principle and thus also serve as the preregistered file versions for stage 2. Code for the power analysis is available via the same OSF link (above). R version 4.3.3 was used for the analyses.

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

## Acknowledgements

This work was supported by the Swedish Research Council for Health, Working Life and Welfare (Forte; 2020-00864). Funders had no role in study design, data collection, analysis, decision to publish, or preparation of the manuscript. We are grateful for comments from reviewers and editors that helped improve the paper.

## Author contributions

D.A.: Idea and study design, analytic strategy, power analysis, data management, data analysis, revised the manuscript; M.L.: Idea and study design, revised the manuscript; G.T.: Idea and study design, revised the manuscript; E.P.: Idea and study design, analytic strategy, drafted the manuscript, revised the manuscript.

## Funding

## Competing interests

M.L. was previously operations manager at one of the regional providers of the healthcare service 1177 direct. All other authors declare no conflicting interests that could have appeared to influence the submitted work.
