## [Transparent Peer Review file · Communications Psychology]

No evidence for decision fatigue using large-scale field data from healthcare

Corresponding Author: Dr Emil Persson

Version 0:

Decision Letter: first round

Dear Dr Persson,

Thank you once again for your manuscript, entitled "A preregistered test of decision fatigue using large-scale field data from healthcare," and for your patience during the peer review process.

Your manuscript has now been evaluated by 2 reviewers, whose comments are included at the end of this letter. Although the reviewers find your protocol to be of interest, they also raise some important concerns. We are very interested in the possibility of proceeding further with your submission in Communications Psychology, but would like to consider your response to these concerns in the form of a revised manuscript before we make a decision on in principle acceptance and Stage 2 submission.

To guide the scope of the revisions, the editors discuss the referee reports in detail within the team, including with the chief editor, with a view to (1) identifying key priorities that should be addressed in revision and (2) overruling referee requests that are deemed beyond the scope of the current study. We hope that you will find the prioritised set of referee points to be useful when revising your study. Please do not hesitate to get in touch if you would like to discuss these issues further.

Most importantly, both reviewers have hesitations regarding the analytical approach. Please revise your manuscript to ensure the analysis allows you to test your hypotheses unambiguously. Where possible, please use your pilot data to verify the approach.

Please also clarify your hypothesis. For Registered Reports, it is of great importance that all hypotheses are unambiguous and that it is specifically stated what statistical outcome they are tied to. Please ensure that the power analysis provides >95% power for your chosen alpha for each hypothesis.

Please address the conceptual points and requests for a better embedding in the literature, including Reviewer #2's point about the link between decision fatigue and mental fatigue.

Finally, please upload the SI (currently called 'Appendix') as a separate document. The entire pilot analysis should be reported in the SI (as is currently the case). The subheadings in the SI need to clarify that each section refers to the pilot analysis.

The protocol, which forms the main manuscript, may not contain Results or Discussion. The Title and Abstract may change at Stage 2.

In sum, we invite you to revise your Stage 1 Registered Report taking into account reviewer and editor comments. Please highlight all changes in the manuscript text file.

In sum, we invite you to revise your Stage 1 Registered Report taking into account reviewer and editor comments. Please highlight all changes in the manuscript text file.

* Include a "Response to reviewers" document detailing, point-by-point, how you addressed each referee comment. If no action was taken to address a point, you must provide a compelling argument. This response will be sent back to the reviewers along with the revised manuscript.

* Ensure that you use our template for Stage 1 Registered Reports to prepare your revised manuscript:

https://www.nature.com/documents/CP_Template-RR-Stage1.docx

Failure to ensure that your revised Stage 1 submission meets our requirements as specified in the template will result in your submission being returned to you, which will delay its consideration.

* In your cover letter, please include the following information:

--An anticipated timeline for completing the study if your Stage 1 submission is accepted in principle.

--A statement confirming that you agree to share your raw data, any digital study materials, computer code, and laboratory log for all eventually published results.

--A statement confirming that, following Stage 1 in principle acceptance, you agree to register your approved protocol on the Open Science Framework (<https://osf.io/>) or other recognised repository, either publicly or under private embargo, until submission of the Stage 2 manuscript.

--A statement confirming that if you later withdraw your paper, you agree to the Journal publishing a short summary of the pre-registered study under a section Withdrawn Registrations.

Link Redacted

We hope to receive your revised manuscript within four to eight weeks. If you cannot send it within this time, please let us know. We will be happy to consider your revision so long as the report still represents a significant contribution to the literature at that stage.

* **TRANSPARENT PEER REVIEW:** Communications Psychology uses a transparent peer review system. This means that we publish the editorial decision letters including Reviewers' comments to the authors and the author rebuttal letters online as a supplementary peer review file. We publish these records for all accepted manuscripts. However, on author request, confidential information and data can be removed from the published reviewer reports and rebuttal letters prior to publication. If your manuscript has been previously reviewed at another journal, those Reviewers' comments would not form part of the published peer review file.

Communications Psychology is committed to improving transparency in authorship. As part of our efforts in this direction, we are now requesting that all authors identified as 'corresponding author' on published papers create and link their Open Researcher and Contributor Identifier (ORCID) with their account on the Manuscript Tracking System (MTS), prior to acceptance. ORCID helps the scientific community achieve unambiguous attribution of all scholarly contributions. You can create and link your ORCID from the home page of the MTS by clicking on 'Modify my Springer Nature account'. For more information please visit www.springernature.com/orcid.

Sincerely,

Antonia Eisenkoeck
Senior Editor
Communications Psychology

REVIEWERS' EXPERTISE:

Reviewer #1: decision fatigue, medical decision making

Reviewer #2: decision fatigue

REVIEWERS' COMMENTS:

Reviewer #1:

Remarks to the Author:

Review of manuscript COMMSPSYCHOL-23-0358-T: A preregistered test of decision fatigue using large-scale field data

from healthcare

The manuscript describes a planned study to use existing healthcare data to empirically test for the existence of decision fatigue in specialist nurses providing triage and advice to patients over the phone. It is submitted for consideration for publication as a Registered Report (Stage 1).

The topic area is an important one as the current literature is limited by a focus to date on retrospective observational designs so pre-registered tests of the decision fatigue phenomenon are warranted and necessary. Many aspects of the proposed study are excellent and I am supportive of the publication of a registered report on this topic. However, I had several reservations about the proposed approach.

1. Hypotheses. The manuscript is framed as a test of whether decision fatigue exists but the hypotheses appear more geared towards exploration of different possible manifestations. The present framing implies that for decision fatigue to exist all hypotheses should be supported. However, some of the predictions are mutually exclusive. H1 (deviation towards personal default rating; essentially regression to own mean) makes sense, but in combination with H2 (deviation towards extreme ratings) suggests changes that are mutually exclusive (unless the personal default is also an extreme rating). Similarly, in H3, the authors propose that nurses will both gather less information (so shorter calls) and process information less efficiently (so longer calls). If all are supported, then would null effects be expected? In addition, I found the logic of H2 difficult. H2 states that with greater decision fatigue both the highest urgency rating and the lowest urgency rating will be chosen more frequently. Based on this phrasing (“and”) I expected this to be tested in one regression (outcome: dummy: 1=extreme rating (highest or lowest), 0=any non-extreme rating). I was also not persuaded that in this context any decision fatigue effect would necessarily move people to the extremes as hypothesised. Rather, I would expect a change relative to a reasonable starting point for the severity of the particular call; i.e. I would not expect that decision fatigue would lead nurses to start sending ambulances to callers reporting moderate symptoms, rather I would expect that they might make somewhat more urgent decisions than they usually do when dealing with moderate symptoms. This could in theory be tested in this dataset.

2. Comparison of AM vs PM workers. While I really liked the quasi-experimental approach and thought it was in principle a very elegant way to test the predictions, I don't think that the groups as currently defined reliably separate out those likely to be decision fatigued from those who are not. “Consider only decisions taken between 13:00–17:00 on Mon–Fri for workers who either take their first call 6:00–9:00 and their last call after 13:00 (AM worker), or take their first call 13:00–18:00 and their last call after 19:00 (PM worker).” This operationalisation means that a call taken by a ‘PM’ worker (who we are assuming will not be decision fatigued during the target window) could feasibly be either their first call taken at 13.00 or their nth call taken at 17.00, hours into a sequence of calls. Consequently, people in these groups are highly likely in theory to be in different stages of decision fatigue and cannot therefore reasonably be grouped and used to compare ‘high’ vs ‘low’ decision fatigue.

3. Analyses. It wasn't clear to me why linear regression is proposed to analyse binary and count outcome variables rather than logistic and Poisson regression. Some additional explanation of this would be helpful. In addition, as there appears to be data available on some of the possible confounders identified in the paper (e.g. call type / severity), could this be included in the analyses?

4. Breaks. It is a real strength that breaks were considered as this is another key limitation in the existing literature. However, breaks are estimated from gaps in calls. This is a reasonable approach, but one which means that non-call tasks which would still constitute work – e.g. paperwork, training etc, could be mistaken for breaks. The likely limitations of this should be acknowledged. Similarly, p10 lines 326-328 –the rules used to identify breaks should be made explicit. It was not clear to me why 8-20 min breaks in the call sequence were identified as true breaks when the standard breaks in the service are described as one 30-min lunch break and two 15-minute breaks.

Minor points

P3, line 59 – it is important to make clear that the debate referred to suggests that it is highly likely that the magnitude of the effect in question was overestimated.

P4, line 103-104 – case assignment is described as essentially random but later evidence is presented that cases show predictable patterns over time “.....likely confounded by time of day effects in composition of call types, which we saw clearly in our data, and call types themselves vary substantially in underlying severity.” This should be clarified.

P5 – the description of RIT arguably overemphasises the conscious rationality of the effect when it has been traditionally conceptualised in the literature as an unconscious bias. It would be beneficial to acknowledge this.

Finally, there is a lack of clarity about whether this is a pre-registered study or a registered report. The study as outlined has already been run as pilot (results reported/known) and what is being registered here is a replication of the same analysis in a different time period of the same dataset. I don't think this is problematic, but the title / framing should transparently reflect the approach and the journal requirements for the chosen format.

Reviewer #2:

Remarks to the Author:

This is a very thorough proposal describing a detailed experimental protocol that has been piloted on extensive preliminary data (>200000 data points). It is well written and carefully thought through.

Despite its undeniable overall quality, there are a few issues that I think still deserve editing :

- The connection of decision fatigue to the general mental fatigue literature comes up late in the manuscript, giving the impression that they are distantly related, when in fact, there is no reason to believe that the mechanisms of decision fatigue would differ in any way from those of mental fatigue. To me, decision fatigue is a specific instance of mental fatigue. Please clarify your take on this point earlier in the introduction.
- Some important, classical confounding factors from the fatigue literature are not discussed : sleepiness, motivation, boredom. It would be worth clarifying how those confounds are accounted for within the proposed design. I think this is fairly obvious but should nevertheless be stated clearly.
- Hypotheses 3 and 4 are presented as being arbitrary, in the sense that "it is of course easy to come up with arguments for why the effect might instead go in the opposite direction". Under those circumstances, they should be withdrawn from the hypothesis list and rather be described as exploratory tests that are going to have no bearing on the conclusion regarding the validity of the decision fatigue concept.
- The choice of binomial dependent variables seems not optimal. Other, more continuous variables, could account for the measures of interest with better sensitivity (e.g. comparing the distributions of urgency ratings).
- Using linear regressions on binomial variables is wrong. If the trials are averaged prior to analysis, this could make linear regressions possible under the condition of sufficient number of trials, but this should then be mentioned, as well as the necessary diagnostic procedures.
- Why running two separate analyses for long vs short break rather than using it as an independent variable?
- Why using OLS rather than mixed models ? A short justification would help here.
- Line 432 : the time indicator variables (=1 for inclusion in relevant 30-min bin) was not described in methods.

Version 1:

Decision Letter: second round

Dear Dr Persson,

Thank you once again for your manuscript, entitled "A strong test of decision fatigue using large-scale field data from healthcare," and for your patience during the peer review process.

Your manuscript has now been evaluated by 2 reviewers, whose comments are included at the end of this letter. We are very interested in the possibility of proceeding further with your submission in Communications Psychology, but would like to consider your response to the one remaining reviewer concern and our editorial requests prior to Stage 1 acceptance in principle. We hope to reach this stage after receiving your revisions and thus it is crucially important that they fully comply with the requests in the checklist.

In response to Reviewer 1, please provide further justification for your choices regarding the classification of breaks.

Editorially, we have provided you a checklist with further requests to guide your revisions (see attachment).

Regarding the interpretation of your effects, you may only interpret either evidence from Bayes Factors or evidence from two-sided equivalence tests as confirmation of no difference/no effect of decision fatigue. We ask you to revise the presentation of your analysis and commitment to interpretation of different patterns of results (which is otherwise well done) in this regard. Should you decide to switch to Bayesian statistics, which is generally very much encouraged, you would need to demonstrate in a new power analysis that the expected sample is sufficiently well-powered as per the guidelines in the checklist and on our webpage,

In sum, we invite you to revise your Stage 1 Registered Report taking into account reviewer and editor comments. Please highlight all changes in the manuscript text file.

* Include a "Response to reviewers" document detailing, point-by-point, how you addressed each referee comment. If no action was taken to address a point, you must provide a compelling argument. This response will be sent back to the reviewers along with the revised manuscript.

* Ensure that you use our template for Stage 1 Registered Reports to prepare your revised manuscript:

https://www.nature.com/documents/CP_Template-RR-Stage1.docx

Failure to ensure that your revised Stage 1 submission meets our requirements as specified in the template will result in your submission being returned to you, which will delay its consideration.

* In your cover letter, please include the following information:

--An anticipated timeline for completing the study if your Stage 1 submission is accepted in principle.

--A statement confirming that you agree to share your raw data, any digital study materials, computer code, and laboratory log for all eventually published results.

--A statement confirming that, following Stage 1 in principle acceptance, you agree to register your approved protocol on the Open Science Framework (<https://osf.io/>) or other recognised repository, either publicly or under private embargo, until submission of the Stage 2 manuscript.

--A statement confirming that if you later withdraw your paper, you agree to the Journal publishing a short summary of the pre-registered study under a section Withdrawn Registrations.

Link Redacted

We hope to receive your revised manuscript within four to eight weeks. If you cannot send it within this time, please let us know. We will be happy to consider your revision so long as the report still represents a significant contribution to the literature at that stage.

* **TRANSPARENT PEER REVIEW:** Communications Psychology uses a transparent peer review system. This means that we publish the editorial decision letters including Reviewers' comments to the authors and the author rebuttal letters online as a supplementary peer review file. We publish these records for all accepted manuscripts. However, on author request, confidential information and data can be removed from the published reviewer reports and rebuttal letters prior to publication. If your manuscript has been previously reviewed at another journal, those Reviewers' comments would not form part of the published peer review file.

Communications Psychology is committed to improving transparency in authorship. As part of our efforts in this direction, we are now requesting that all authors identified as 'corresponding author' on published papers create and link their Open Researcher and Contributor Identifier (ORCID) with their account on the Manuscript Tracking System (MTS), prior to acceptance. ORCID helps the scientific community achieve unambiguous attribution of all scholarly contributions. You can create and link your ORCID from the home page of the MTS by clicking on 'Modify my Springer Nature account'. For more information please visit www.springernature.com/orcid.

Sincerely,

Jennifer Bellingtier, PhD
Senior Editor
Communications Psychology

REVIEWERS' EXPERTISE:

Reviewer #1: decision fatigue, medical decision making

Reviewer #2: decision fatigue

REVIEWERS' COMMENTS:

Reviewer #1:

Remarks to the Author:

The authors have provided a thorough and measured response to the points that we raised. In particular, we felt that; the revised hypotheses more clearly align with the predictions made; the additional clarification around the appropriateness of the AM/PM quasi-experimental grouping is reassuring; the move to logistic regression for analyses of binary outcomes is welcome; and the re-framing of the study as a registered confirmatory test is more transparent. All minor issues were also addressed very well.

The one area of the response we would query is the decision to classify breaks as 8-60 mins off-call. While the authors are more familiar with this clinical setting / dataset than we are, including such a wide interval in a service where the longest standard break is known a priori to be 30 minutes would seem to increase the probability of off-call work tasks being misclassified as breaks. We would suggest that 8-40 minutes seems a more appropriate interval but defer to the author's knowledge of what is most appropriate for this setting.

Julia Allan, University of Stirling & Mona Maier, University of Aberdeen

Reviewer #2:

Remarks to the Author:

The authors have taken my comments carefully into account. I am fully satisfied with the new version of the manuscript.

Version 2:

Decision Letter: third round

7th May 2024

Dear Dr Persson,

Thank you once again for submitting your revised Stage 1 Registered Report, entitled "A strong test of decision fatigue using large-scale field data from healthcare." Everything is in order and I am delighted to say that we can offer acceptance in principle. You may progress to Stage 2 and complete the study as approved.

As you know, a condition of in-principle-acceptance is that the authors agree to deposit their Stage 1 accepted protocol in a repository, either publicly or under embargo until Stage 2 acceptance and publication. We are very keen to showcase our in-principle accepted protocols, so that our readers, reviewers, and potential authors can gain insight into the requirements of the format as well as an idea of the types of projects that are suitable for publication in Communications Psychology. We have set up a space on figshare to host all of our in-principle accepted protocols, which can either be made public or kept under embargo until Stage 2 acceptance (depending on author preference). This gives you the opportunity to have your work publicly associated with Communications Psychology, and of course we will be very pleased to showcase your report if you agree to share it publicly.

Depositing the work on our figshare space does not preclude deposition of your Stage 1 protocol on other depositories – your protocol can also be posted on OSF, Dataverse, Dryad or any other public repository of your choice. You also do not need to do anything – if you agree with posting your protocol on our figshare space, we will upload your protocol on your behalf and either set it public or place it under embargo, depending on your choice. Your protocol will be licensed under a CC BY license (Creative Commons Attribution 4.0 International License). The CC BY license allows for maximum dissemination and re-use of open access materials and is preferred by many research funding bodies. Under this license users are free to share (copy, distribute and transmit) and remix (adapt) the contribution including for commercial purposes, providing they attribute the contribution in the manner specified by the author or licensor (read full legal code: <http://creativecommons.org/licenses/by/4.0/legalcode>) Please note that any use of <https://springernature.figshare.com> will be subject to the Figshare terms of use. Figshare has the right to enforce these terms and conditions where applicable. Use of third party services and sites will be subject to the relevant terms of use and will apply if we act on your behalf in this regard. Do let me know if you would like to take up this option or if you have any questions regarding the protocol deposition requirement.

IMPORTANT:

In cases where the registered experimental design is altered after AIP due to unforeseen circumstances (e.g. change of equipment or unanticipated technical error), the authors should consult the editors immediately for advice, prior to the completion of data collection.

Following completion of your study, we invite you to resubmit your paper for peer review as a Stage 2 Registered Report. Please note that your manuscript can still be rejected for publication at Stage 2 if the Editors consider any of the following to hold:

- The results were unable to test the authors' proposed hypotheses by failing to meet the approved outcome-neutral criteria
- The authors altered the Introduction, rationale, or hypotheses, as approved in the Stage 1 submission
- The authors failed to adhere closely to the registered experimental procedures without previously seeking editorial approval
- Any post hoc (unregistered) analyses were either unjustified, insufficiently caveated, or overly dominant in shaping the authors' conclusions
- The authors' conclusions were not justified given the data obtained

We encourage you to read the complete guidelines for authors concerning Stage 2 submissions at <https://www.nature.com/commspsychol/submit/registered-reports> and <https://www.nature.com/documents/commspsychol-style-formatting-checklist-article-rr.pdf>.

Please especially note the requirements for protocol deposition, data sharing, and that withdrawing your manuscript will result in publication of a Retracted Registration.

When you are ready, please use the following link to access your home page and submit your Stage 2 Registered Report:

Link Redacted

*This url links to your confidential homepage and associated information about manuscripts you may have submitted or be reviewing for us. If you wish to forward this e-mail to co-authors, please delete this link to your homepage first.

* **TRANSPARENT PEER REVIEW:** Communications Psychology uses a transparent peer review system. This means that we publish the editorial decision letters including Reviewers' comments to the authors and the author rebuttal letters online as a supplementary peer review file. This means that the records will be published together with your Stage 2 report. On author request, confidential information and data can be removed from the published reviewer reports and rebuttal letters prior to publication. If your manuscript has been previously reviewed at another journal, those Reviewers' comments would not form part of the published peer review file.

We expect your Stage 2 Registered Report to be submitted by the date specified in your latest cover letter. If unforeseen circumstances prevent submission by that date, please contact us as soon as possible to discuss any changes to the submission time-frame.

Thank you again for offering us this work and we look forward to receiving your Stage 2 Registered Report.

Yours sincerely,

Jennifer Bellingtier, PhD
Senior Editor
Communications Psychology

Version 3: fourth round

Decision Letter:

Dear Dr Persson,

Thank you for submitting your manuscript titled "A strong test of decision fatigue using large-scale field data from healthcare" to Communications Psychology. We have given the paper our careful consideration and find it of potential interest. However, due to certain shortcomings we are concerned that sending the current manuscript out to review could lead to unnecessary delays and quite possibly an undesirable outcome of the review process.

In particular:

1. Please add a sensitivity analysis for the achieved power for the targeted effect size (not the effect size you found) at the achieved sample.
2. Each deviation in the Introduction and Method there must be accompanied by a justification [in square brackets] that provides an explanation for the change, except for changes in tense.
3. Please do not delete information about the Pilot data. For example, the sentence, "Applied to the target data (stage 2), using these rules resulted in a final sample size of $n = 9,648$ calls to $k = 143$ nurses" should instead read, "Applied to the pilot data these results resulted in a final sample size of $n = XX$ calls to $k = X$ nurses; applied to the target data (stage 2), using these rules resulted in a final sample size of $n = 9,648$ calls to $k = 143$ nurses".

We would therefore like to invite you to revise your manuscript to address these concerns before we make a final determination on whether to send your manuscript for external review.

We shall hope to receive your revised version as soon as you are able to complete the suggested revisions. If something similar is published in the interim we will have to consider the impact it has on the novelty of a revised manuscript.

If you anticipate a delay of more than four weeks, please let us know. Should your manuscript be substantially delayed without notifying us in advance and your article is eventually published, the received date may be that of the revised, not the original, version.

If you are not interested in submitting a suitably revised manuscript in the future please let me know immediately so we can close your file. If you have any questions, please contact me.

Please use the link below when you are prepared to resubmit.
Link Redacted

Thank you for your interest in Communications Psychology.

Best regards,
Jennifer Bellingtier

Jennifer Bellingtier, PhD
Senior Editor
Communications Psychology

Version 4: fifth round

Decision Letter:

8th Jan 2025

Dear Dr Persson,

Thank you once again for submitting your Stage 2 Registered Report, entitled "A strong test of decision fatigue using large-scale field data from healthcare," and for your patience during the re-review process.

Your manuscript has now been evaluated by Reviewer 1 from the previous rounds of review, whose comments are included at the end of this letter. In the light of our reviewers' advice, we are pleased to inform you that we will be able to accept your Stage 2 manuscript, pending revisions to address Reviewers' comments and editorial requests.

To guide the scope of the revisions, the editors discuss the referee reports in detail within the team, including with the chief editor, with a view to (1) identifying key priorities that should be addressed in revision and (2) overruling referee requests that are beyond the scope of Stage 2 Registered Reports.

We encourage you to address the discussion points raised by Reviewer 1. There should be no further analyses completed at this stage.

One of the main reasons for delays in eventual acceptance is failure to fully comply with editorial policies and formatting requirements. To assist you with finalizing your manuscript for publication, I attach our Editorial Requests Table which lists all of our editorial policies and formatting requirements. I have also attached a marked-up copy of the article file with additional guidance.

Please attend to *every item* in the Table and upload a copy of the completed checklist with your submission. I have highlighted in the checklist items that require your attention. I also mention here a few points that are frequently missed and can cause delays:

- 1) Ensure that all corresponding authors have linked their ORCID to their account on our online manuscript handling system. This is very frequently missed and invariably causes delays in formal acceptance.
- 2) Ensure that you provide all of the materials requested in the attached checklist and below with your final submission.

OPEN ACCESS:

Communications Psychology is a fully open access journal. Articles are made freely accessible on publication. For further information about article processing charges, open access funding, and advice and support from Nature Research, please visit <https://www.nature.com/commpsychol/open-access>

* **CODE AVAILABILITY:** To proceed to formal acceptance, you must now publicly deposit the custom analysis code supporting your conclusions; please use a repository that mints the code with a digital object identifier (DOI). The manuscript must include a section titled "Code Availability" at the end of the methods section. The link to the repository and the DOI

must be included in the Code Availability statement. Publication as Supplementary Information will not suffice.

*** DATA AVAILABILITY:**

It is a requirement for Registered Reports to make data publicly available. All Communications Psychology manuscripts must include a section titled "Data Availability" at the end of the Methods section. More information on this policy, is available in the Editorial Requests Table and at <http://www.nature.com/authors/policies/data/data-availability-statements-data-citations.pdf>. Please share a link to your publicly deposited data in the Data Availability statement.

We hope to hear from you within three weeks; please let us know if the revision process is likely to take longer.

Please use the following link for uploading the materials:
Link Redacted

With best regards,

Jennifer Bellingier, PhD
Senior Editor
Communications Psychology

Reviewer #1:

Remarks to the Author:

This is an elegantly designed, interesting and well conducted study that advances the literature on decision fatigue.

- The data obtained are suitable to test the authors' proposed hypotheses.
- The introduction, rationale and hypotheses are the same as the approved Stage 1 submission.
- The authors adhered to the registered experimental procedures.
- The only additional (non-prespecified) content added is relevant and clearly justified (sensitivity analyses when sample size was smaller than anticipated).
- The conclusions are justified given the data.

I have 5 queries and suggestions for the authors.

(1) The proportion of calls assigned a more urgent rating increases over time. The authors argue on p27 (and I agree) that this may reflect confounding from temporal patterns in call composition. They refer to supplementary materials in the Stage 1 report for further detail which I unfortunately cannot access at the time of writing. However, "a similar pattern emerged for calls taken later during a work shift as indexed by call order". It is unclear to me why this measure would be similarly confounded as the authors state that "There is substantial variation in scheduling within individual nurses. Morning, afternoon, and evening shifts come in different configurations, with staggered start times and varying length". If shift start and finish times are highly varied, why would call order (i.e. within an individual nurses' shift) be confounded by temporal patterns in call type over the day? This is worth clarifying in the main text of the manuscript.

(2) On p37 the authors state that "From a practical perspective these findings are reassuring. An implication for policymaking is that generic concerns about decision fatigue should have little influence on policy in healthcare or elsewhere." I would recommend amending to acknowledge that this study raises additional questions with practical/policy relevance. The results make a persuasive case that decision fatigue may not be the mechanism which explains observed changes in health professional decision making over the working day. However, even if the patterns observed in the wider literature result from (suggested confounding) factors such as boredom or lack of motivation, this still has policy relevance.

(3) On p37 the authors state that "Higher urgency ratings were assigned more often later in the day. It would have been easy to build a case (in good faith) where this pattern was taken as prima facie evidence of decision fatigue.... However, that interpretation would likely be wrong because the observed association between time of day and assigned urgency is likely confounded by other time of day dependent effects, for example composition of call types". Some additional discussion at this point would be welcome as this will be a key point for the field going forward. There are examples of studies showing changes which align with decision fatigue from e.g. experimentally controlled studies where possible confounders are controlled (e.g. *Frontiers | Overcoming Therapeutic Inertia in Multiple Sclerosis Care: A Pilot Randomized Trial Applying the Traffic Light System in Medical Education*), from contexts where the decision in question/context is highly uniform (e.g. decisions to wash hands on entering a ward <https://pubmed.ncbi.nlm.nih.gov/25365728/>; scan reading, etc) and from contexts where the decisions under study are e.g. prescribing particular drugs for particular conditions (<https://journals.sagepub.com/doi/full/10.1177/0272989X241263823>) and where there is no obvious reason why one might expect that the number of patients presenting with osteoporosis, or deemed suitable for a statin for example would vary systematically across the day. This seems worth some additional discussion.

(4) On p39 where the authors state ".....other organization-specific factors like training or guidelines might play a role", it

would be good to specify at this point whether decision support algorithms are used in this setting, and if they are used, the possible impact on the study results. In the comparable UK service, nurses utilise decision support software that scaffolds the decisions made. If such decision support systems are in place, this may have relevance to both the effort involved in decision making and to the variation in outcomes that is possible within each consultation.

(5) Finally, I would recommend aligning the phrasing of the abstract with the discussion/conclusion. In the discussion, the authors write that “the relationship between fatigue and information processing or cognitive effort avoidance could be more subtle or complicated than previously thought. It might be the case that there is a real effect that is more nuanced, weaker, potentially heterogeneous or context dependent”. In the conclusion, they write “Whereas these findings don’t preclude the existence of a weaker version of decision fatigue or more context specific effects, they cast serious doubt on the empirical relevance of decision fatigue as a domain general effect for sequential decisions in healthcare and elsewhere”. This is a measured way to summarise the results which encourages further investigation of possible mechanisms. It would be good if the abstract wording was consistent with this.

Minor point

p27: Please rephrase the sentence “Proportion consult ranged from 13.8% to 22.7%.” to make clear to the reader exactly what this refers to.

Thank you for the opportunity to review this paper. It’s impressive work which has given me a great deal of food for thought.

Julia Allan

The images or other third party material in this Peer Review File are included in the article’s Creative Commons license, unless indicated otherwise in a credit line to the material. If material is not included in the article’s Creative Commons license and your intended use is not permitted by statutory regulation or exceeds the permitted use, you will need to obtain permission directly from the copyright holder.

Reviewer #1:

Remarks to the Author:

Review of manuscript COMMSPSYCHOL-23-0358-T: A preregistered test of decision fatigue using large-scale field data from healthcare

Comment. *The manuscript describes a planned study to use existing healthcare data to empirically test for the existence of decision fatigue in specialist nurses providing triage and advice to patients over the phone. It is submitted for consideration for publication as a Registered Report (Stage 1). The topic area is an important one as the current literature is limited by a focus to date on retrospective observational designs so pre-registered tests of the decision fatigue phenomenon are warranted and necessary. Many aspects of the proposed study are excellent and I am supportive of the publication of a registered report on this topic. However, I had several reservations about the proposed approach.*

Response. Thank you for your positive evaluation of our paper.

Comment 1. Hypotheses. *The manuscript is framed as a test of whether decision fatigue exists but the hypotheses appear more geared towards exploration of different possible manifestations. The present framing implies that for decision fatigue to exist all hypotheses should be supported. However, some of the predictions are mutually exclusive. H1 (deviation towards personal default rating; essentially regression to own mean) makes sense, but in combination with H2 (deviation towards extreme ratings) suggests changes that are mutually exclusive (unless the personal default is also an extreme rating). Similarly, in H3, the authors propose that nurses will both gather less information (so shorter calls) and process information less efficiently (so longer calls). If all are supported, then would null effects be expected? In addition, I found the logic of H2 difficult. H2 states that with greater decision fatigue both the highest urgency rating and the lowest urgency rating will be chosen more frequently. Based on this phrasing (“and”) I expected this to be tested in one regression (outcome: dummy: 1=extreme rating (highest or lowest), 0=any non-extreme rating). I was also not persuaded that in this context any decision fatigue effect would necessarily move people to the extremes as hypothesised. Rather, I would expect a change relative to a reasonable starting point for the severity of the particular call; i.e. I would not expect that decision fatigue would lead nurses to start sending ambulances to callers reporting moderate symptoms, rather I would expect that they might make somewhat more urgent decisions than they usually do when dealing with moderate symptoms. This could in theory be tested in this dataset.*

Response. Thank you. We have now clarified our hypotheses and their link to our overall test of decision fatigue. In particular, following your suggestion we reframed H2 as an expected gradual increase in urgency ratings, and we withdrew H3 (and H4) due to the potential ambiguity about expected direction for effects that would be diagnostic of decision fatigue. We may still conduct exploratory tests of these variables (call duration and consultancy) but they will have no bearing on the overall conclusion regarding the validity of the decision fatigue concept. (Withdrawing H3 and H4 and instead treating them as exploratory tests was also suggested by another referee.) We have also clarified how the planned tests of our hypotheses (now H1 and H2) unambiguously map to an overall conclusion of *support for* or *against* decision fatigue. Overall, we believe that these changes have substantially improved our protocol.

Implementing these changes led to edits on several places in the manuscript, including the *Introduction*, where the hypotheses are described and motivated, the *Evaluation* subsection (under *Methods/Analysis Plan*), where the updated plan for evaluation is described, and the overall summary presented in the *Design table* (Table 1, at the end of the submitted manuscript).

Comment 2. Comparison of AM vs PM workers. While I really liked the quasi-experimental approach and thought it was in principle a very elegant way to test the predictions, I don't think that the groups as currently defined reliably separate out those likely to be decision fatigued from those who are not. "Consider only decisions taken between 13:00–17:00 on Mon–Fri for workers who either take their first call 6:00– 9:00 and their last call after 13:00 (AM worker), or take their first call 13:00–18:00 and their last call after 19:00 (PM worker)." This operationalisation means that a call taken by a 'PM' worker (who we are assuming will not be decision fatigued during the target window) could feasibly be either their first call taken at 13.00 or their nth call taken at 17.00, hours into a sequence of calls. Consequently, people in these groups are highly like in theory to be in different stages of decision fatigue and cannot therefore reasonably be grouped and used to compare 'high' vs 'low' decision fatigue.

Response. We agree that in theory workers could be in different stages of decision fatigue and that the design admits the possibility that a call to a 'PM' worker is either their first or on the back of a number of calls already handled. However, while such cases are feasible, they are rare in practice; overall there is substantial separation of cumulative workload between AM and PM workers during this time window. In the pilot data, AM workers had on average spent 6.5 hours (SD = 0.8) at work and handled 26.9 calls compared to 0.6 hours (SD = 0.6) and 4.6 calls for PM workers. Thus, whereas other designs could be considered, we believe that the binary grouping that we suggest is feasible, straightforward, and has the clear advantage that other potential confounds are well accounted for while there is on average a substantial difference in cumulative workload between the two groups at the point of evaluation.

However, re-reading our manuscript, we realize that we could have made these points clearer when describing this comparison. Thus, we have now included part of the description from the pilot data also in the main protocol (manuscript), at the end of the *Identifying best comparison points ...* subsection (under *Methods/Design*).

The following text (and corresponding figure) was added:

"Fig. 2 shows the substantial separation in cumulative workload between morning and afternoon workers that took calls inside the same time window. Morning workers had on average spent 6.5 hours (SD = 0.8, n = 6,221 calls) at work and handled 26.9 calls (SD = 9.1, n = 6,221 calls) compared to 0.6 hours (SD = 0.6, n = 6,451 calls) and 4.6 calls (SD = 3.6, n = 6,451 calls) for afternoon workers. Thus, if there are relevant effects from fatigue on decision-making and judgment, as the concept decision fatigue presupposes, we should arguably see them here.

Fig. 2. Separation in cumulative workload for observations included in the AM-PM shift overlap subsample. **Left:** Distribution of time spent at work by type of shift (AM vs. PM) for calls handled during the overlap time

window. **Right:** Distribution of number of previous calls on the same day by type of shift (AM vs. PM) for calls handled during the overlap time window. See Data inclusion and exclusion subsection below for details on how the AM-PM shift overlap subsample was selected. (Data source: pilot data.)”

Comment 3. *Analyses. It wasn't clear to me why linear regression is proposed to analyse binary and count outcome variables rather than logistic and Poisson regression. Some additional explanation of this would be helpful. In addition, as there appears to be data available on some of the possible confounders identified in the paper (e.g. call type / severity), could this be included in the analyses?*

Response. We have followed your suggestion and changed to logistic regression for analysis of binary variables. Moreover, per suggestion by another referee we switched to a mixed-models framework. In brief, with these changes, our main analytical approach is now to test statistically for effects consistent with decision fatigue for each dependent variable (each hypothesis) in both subsamples using a mixed model for binary or continuous dependent variables (*Default* and *Urgency*, respectively), with random intercepts and nurse-level fixed effects and clustered standard errors. As before, time indicator variables (=1 for inclusion in relevant 30-min bin) are entered as fixed factors in the overlap subsample. For more details, please see the *Main confirmatory tests* subsection (under *Methods/Analysis Plan*) in the manuscript.

We also considered the very reasonable suggestion to include additional data on some of the possible confounders identified in the paper (e.g., call type / severity). However, a potential downside with such an approach is that the call type categorization is both extensive (ca 200 categories) and unevenly distributed, with a minority of large categories (i.e., many calls) and a non-negligible amount of small categories that receive a low number of calls. Thus, including this variable as a control factor without modifying or aggregating it would likely not improve the models and would not seem like an efficient way to estimate our effects of interest. Of course, it is possible that this variable could be aggregated into fewer categories, but there is no obvious or straightforward way to do this, and we would worry about introducing bias. For this reason, our preferred approach is to not include these additional data in the analyses.

Comment 4. *Breaks. It is a real strength that breaks were considered as this is another key limitation in the existing literature. However, breaks are estimated from gaps in calls. This is a reasonable approach, but one which means that non-call tasks which would still constitute work – e.g. paperwork, training etc, could be mistaken for breaks. The likely limitations of this should be acknowledged. Similarly, p10 lines 326-328 –the rules used to identify breaks should be made explicit. It was not clear to me why 8-20 min breaks in the call sequence were identified as true breaks when the standard breaks in the service are described as one 30-min lunch break and two 15-minute breaks.*

Response. Thank you for emphasizing that breaks are important to study here. We agree. We have followed your suggestion to acknowledge as a limitation that non-call tasks could be mistaken for breaks. Thus, in the *Data inclusion and exclusion* subsection (under *Methods/Sampling Plan*), at the end of the second paragraph, we added: “This is a suitable approach, but also note a limitation for the analysis of calls around breaks, since non-call tasks, which would still constitute work, e.g., paperwork, training, could be mistaken for breaks.”

We also clarified why we chose the exact values to identify breaks (e.g., previously 8-20 min for short breaks, which we changed – see below). There is nothing specific about these exact values, but we wanted to make sure to specify them exactly to credibly restrict analytic flexibility later on. Moreover, noting (in the pilot data) that there was substantial variation in observed time difference between calls (which we use to identify what are likely true breaks) in the region ca >5 min to around 60 min, which is plausible given the flexible work setting (e.g., overlapping schedule and break times

between workers), we wanted to use reasonably wide intervals when identifying breaks to avoid losing too many observations. To clarify our approach, in the manuscript (*Data inclusion and exclusion* subsection, at the end of the breaks paragraph), we added: “The exact values used here are arbitrary (unless changed substantially), e.g., using 7 or 9 min instead of 8 min as lower cutoff for a break should not make a difference; the important part for us is to decide on and specify these values *exactly and in advance*, in order to credibly restrict analytic flexibility later on when analyzing the new data.”

Finally, please note that we decided to merge the short and long breaks samples, and thus the cutoffs for identifying breaks changed (now 8–60 min). (This approach was suggested by another reviewer.)

Minor points

Comment 5. P3, line 59 – *it is important to make clear that the debate referred to suggests that it highly likely that the magnitude of the effect in question was overestimated.*

Response. We have done as suggested. In the relevant paragraph, we added (here shown in *italics*): “These results spurred much debate²²⁻²⁵, *primarily arguing that the magnitude of the effect was likely overestimated.* There was a conceptual replication²⁶ (successful but *indicating a weaker effect*, to date not published), and ...”

Comment 6. P4, line 103-104 – *case assignment is described as essentially random but later evidence is presented that cases show predictable patterns over time “.....likely confounded by time of day effects in composition of call types, which we saw clearly in our data, and call types themselves vary substantially in underlying severity.” This should be clarified.*

Response. We agree this should be clarified. Thank you for noting this. What we meant here is that workers answer to incoming calls without knowing who is calling or why, since when they are ready to answer the computer simply assigns them the first call in line waiting. This prevents deliberate sorting of cases by patient characteristics, which could for example be an issue if looking at other types of medical data where decisions or procedures can be planned (scheduled) in advance. However, time of day is still a potential confounder since patients with different types of medical problems could be calling at different times of day, and this is what we refer to in the passage quoted in the comment above. In our design we handle this potential problem by looking at cases that are reasonably close in time.

We have now clarified this in the manuscript. In the *Introduction*, the relevant sentence now reads (added text shown in *italics*): “In our setting this is not a problem because case assignment is essentially random *at a given time point, among workers*, which rules out *deliberate* sorting of cases (*scheduling*) based on prior knowledge of patient type or characteristics.”

Comment 7. P5 – *the description of RIT arguably overemphasises the conscious rationality of the effect when it has been traditionally conceptualised in the literature as an unconscious bias. It would be beneficial to acknowledge this.*

Response. We have done as suggested. This is now acknowledged in the *Introduction* (toward the end), where we write (text shown in *italics* is new): “It is important to note that these hypotheses and the underlying behavioral theory represent one, but not the only, way to conceptualize decision

fatigue. In our view it is a coherent and the most sensible approach for our setting, *although the description (and application of rational inattention theory) used here arguably overemphasizes the conscious rationality of the effect.*"

Comment 8. *Finally, there is a lack of clarity about whether this is a pre-registered study or a registered report. The study as outlined has already been run as pilot (results reported/known) and what is being registered here is a replication of the same analysis in a different time period of the same dataset. I don't think this is problematic, but the title / framing should transparently reflect the approach and the journal requirements for the chosen format.*

Response. Thank you for pointing this out. We have now clarified that this is indeed a registered report and describe more clearly what parts are registered. This led to edits on several places in the manuscript and associated supplement. The most important, topical edits that were made are that we no longer use the term "preregistered" (e.g., the title is now: "Stage 1 Registered Report: A strong test of decision fatigue using large-scale field data from healthcare"), and that we now consistently describe our approach of the form "explored pilot data ... and with this protocol we now plan a confirmatory test of the decision fatigue hypothesis using new data" (for exact formulations, see in particular the end of the first paragraph in the *Introduction* or the description in the *Summary of approach* subsection under the *Methods/Design*). We also removed the appendix (with pilot data) from the main protocol and now instead submitted it as a standalone supplementary file, where we also more clearly marked that this indeed constitutes analysis of pilot data and as such will not be part of our planned final, confirmatory, analysis for this project.

Thank you for reading our paper in detail and providing comments and suggestions that substantially improved it.

Reviewer #2:

Remarks to the Author:

Comment. *This is a very thorough proposal describing a detailed experimental protocol that has been piloted on extensive preliminary data (>200000 data points). It is well written and carefully thought through.*

Response. Thank you for your positive evaluation of our paper.

Despite its undeniable overall quality, there are a few issues that I think still deserve editing:

Comment 1. *The connection of decision fatigue to the general mental fatigue literature comes up late in the manuscript, giving the impression that they are distantly related, when in fact, there is no reason to believe that the mechanisms of decision fatigue would differ in any way from those of mental fatigue. To me, decision fatigue is a specific instance of mental fatigue. Please clarify your take on this point earlier in the introduction.*

Response. We agree on this interpretation (that decision fatigue can be seen as a specific instance of mental fatigue) and we have now made this connection clearer in the paper.

In the *Introduction*, after the first sentence (which introduces the decision fatigue concept), we added: "It can be seen as a specific instance of the more general notion mental fatigue⁵⁻⁸, with overlapping proximal causes but the emphasis on decision-making."

Also in the *Introduction*, further down, at the end of the paragraph describing the conceptual basis of decision fatigue, we added: "Taken together, this conceptualization is consistent with the idea that decision fatigue can be seen as a specific instance of mental fatigue, which presupposes that prolonged cognitive activity (in this case primarily decision-making and hard thinking) may lead to aversive feelings and tiredness and subsequently a willingness to disengage⁵⁻⁸."

References

[5] Dora, J., van Hooff, M. L. M., Geurts, S. A. E., Kompier, M. A. J. & Bijleveld, E. The effect of opportunity costs on mental fatigue in labor/leisure trade-offs. *J. Exp. Psychol. Gen.* 151, 695-710, doi: 10.1037/xge0001095 (2022).

[6] Wang, Z., Chang, Y., Schmeichel, B. J. & Garcia, A. A. The effects of mental fatigue on effort allocation: modeling and estimation. *Psychol. Rev.*, doi: 10.1037/rev0000365 (2022).

[7] Hockey, R. *The Psychology of Fatigue: Work, Effort and Control*. (Cambridge University Press, 2013).

[8] Inzlicht, M., Schmeichel, B., & Macrae, C. N. Why self-control seems (but may not be) limited. *Trends Cogn. Sci.* 18, 127-133, doi: 10.1016/j.tics.2013.12.009 (2014).

Comment 2. *Some important, classical confounding factors from the fatigue literature are not discussed: sleepiness, motivation, boredom. It would be worth clarifying how those confounds are accounted for within the proposed design. I think this is fairly obvious but should nevertheless be stated clearly.*

Response. Thank you for this suggestion. We have now added a separate subsection in *Methods/Design* entitled “Accounting for some classical confounds in the fatigue literature.” The added text in full reads:

“As described above, the chosen subsamples (breaks and overlap) provide good comparison points because the pseudorandom allocation of calls to workers prevents scheduling by case difficulty, and the temporal closeness of comparison points means that we avoid potential confounding due to time of day variations in calls. This design also accounts for classical confounding factors discussed in the fatigue literature, including boredom, sleepiness, and motivation. Here the reasoning is that boredom or sleepiness, or lack of motivation, in a low-demand condition could mask real effects from decision fatigue (induced by a comparable high-demand condition), because they may lead to disengagement and as a result similar behavioral effects as are expected from decision fatigue⁶⁵. In our design such potential confound is unlikely, because workers at low-fatigue comparison points (e.g., afternoon shift during the overlap time window) are performing the same type of qualified work, just at lower cumulative intensity; and external factors like type of calls are approximately similar. Sleepiness is similarly accounted for, since comparisons are always close in time, although some caution might be warranted for analysis of the breaks subsample during night, where it is possible that sleepiness is more pronounced going into a break than right after.

There is also a flipside argument about motivation, which, when high and under the right circumstances, seems to help people overcome fatigue even after bouts of cognitive activity⁶⁶. Tasks that are inherently fun or excessively rewarding, and where people have full situational control over how much effort to exert, are thus according to this view unlikely to induce fatigue on a substantial level⁶⁷. However, at work people often do not have full situational control to adjust their effort seamlessly; there are expectations to perform to a certain level, and if demand is high, hard work is likely needed. From this perspective, using archival/field data from settings where work is demanding, repetitive, and requiring qualified assessments on back-to-back cases, like in our case, seems suitable for testing decision fatigue.”

References

[65] Milyavskaya, M., Inzlicht, M., Johnson, T. & Larson, M. J. Reward sensitivity following boredom and cognitive effort: A high-powered neurophysiological investigation. *Neuropsychologia* 123, 159-168, doi:10.1016/j.neuropsychologia.2018.03.033 (2019).

[66] Boksem, M. A., Meijman, T. F. & Lorist, M. M. Mental fatigue, motivation and action monitoring. *Biol. Psychol.* 72, 123-132, doi:10.1016/j.biopsycho.2005.08.007 (2006).

[67] Boksem, M. A. & Tops, M. Mental fatigue: costs and benefits. *Brain research reviews* 59, 125-139, doi:10.1016/j.brainresrev.2008.07.001 (2008).

Comment 3. *Hypotheses 3 and 4 are presented as being arbitrary, in the sense that “it is of course easy to come up with arguments for why the effect might instead go in the opposite direction”. Under those circumstances, they should be withdrawn from the hypothesis list and rather be described as exploratory tests that are going to have no bearing on the conclusion regarding the validity of the decision fatigue concept.*

Response. We have done as suggested (withdrew H3 and H4 but we may still conduct exploratory tests of these variables, i.e., call duration and consultancy, but they will have no bearing on the overall conclusion regarding the validity of the decision fatigue concept).

Comment 4. *The choice of binomial dependent variables seems not optimal. Other, more continuous variables, could account for the measures of interest with better sensitivity (e.g. comparing the distributions of urgency ratings).*

Response. We have followed this advice for H2, which we also reframed as an expected gradual increase in urgency ratings, using average urgency ratings as the associated, more continuous, main outcome variable.

Comment 5. *Using linear regressions on binomial variables is wrong. If the trials are averaged prior to analysis, this could make linear regressions possible under the condition of sufficient number of trials, but this should then be mentioned, as well as the necessary diagnostic procedures.*

Response. We changed to logistic regression for binomial dependent variables.

Comment 6. *Why running two separate analyses for long vs short break rather than using it as an independent variable?*

Response. We now merged the short and long breaks samples as suggested. (We further used the pilot data to determine whether we should include break length as an independent control variable, but model specification tests rejected this in favor of the more parsimonious model.)

Comment 7. *Why using OLS rather than mixed models? A short justification would help here.*

Response. We initially used standard OLS primarily because it is a straightforward analytical approach (including easy to interpret) that is generally considered robust for estimating average treatment effects. We did not consider mixed models at the outset, but, upon reflection, we like the idea and it seems suitable for our setting and design. We therefore switched to a mixed-models approach, treating individual intercepts as random factor and time indicator variables and workers as fixed in the overlap sample, and, similarly, used random-factor individual intercepts in the breaks sample. For more details on specification, please see the *Main confirmatory tests* subsection (under *Methods/Analysis Plan*) in the manuscript.

Comment 8. *Line 432: the time indicator variables (=1 for inclusion in relevant 30-min bin) was not described in methods.*

Response. We now describe these variables in *Methods*. In the *Data inclusion and exclusion* subsection (under *Methods/Sampling plan*) we added the following text to the description of the overlap sample: “Further, we divide the relevant time window (13:00–17:00) in 30-min time bins (thus eight in total) and keep only those bins with >100 observations for each category of workers (AM and PM shifts, respectively).” Then, in the *Independent variables* subsection (under

Methods/Analysis Plan), we added (at the end): “In addition, we define time indicator variables for inclusion (=1) in relevant 30-min time bins inside the overlap time window. For example, Time 1 indicates whether or not a particular call occurred between 13:00 and 13:30 (in the overlap sample), and so on. We include these as independent control variables to account for potential variations over time inside the overlap time window.”

Thank you for reading our paper in detail and providing comments and suggestions that substantially improved it.

Reviewer #1:

Remarks to the Author:

Comment. *The authors have provided a thorough and measured response to the points that we raised. In particular, we felt that; the revised hypotheses more clearly align with the predictions made; the additional clarification around the appropriateness of the AM/PM quasi-experimental grouping is reassuring; the move to logistic regression for analyses of binary outcomes is welcome; and the re-framing of the study as a registered confirmatory test is more transparent. All minor issues were also addressed very well.*

The one area of the response we would query is the decision to classify breaks as 8-60 mins off-call. While the authors are more familiar with this clinical setting / dataset than we are, including such a wide interval in a service where the longest standard break is known a priori to be 30 minutes would seem to increase the probability of off-call work tasks being misclassified as breaks. We would suggest that 8-40 minutes seems a more appropriate interval but defer to the author's knowledge of what is most appropriate for this setting.

Response. We have followed your advice and changed to 8-40 mins classification.

Thank you again for reviewing our manuscript and providing comments and suggestions that helped improving it.

Response to Reviewer #1

Comment. *This is an elegantly designed, interesting and well conducted study that advances the literature on decision fatigue. The data obtained are suitable to test the authors' proposed hypotheses. The introduction, rationale and hypotheses are the same as the approved Stage 1 submission. The authors adhered to the registered experimental procedures. The only additional (non-prespecified) content added is relevant and clearly justified (sensitivity analyses when sample size was smaller than anticipated). The conclusions are justified given the data.*

Response: Thank you for this positive evaluation of our paper.

I have 5 queries and suggestions for the authors.

Comment 1. *(1) The proportion of calls assigned a more urgent rating increases over time. The authors argue on p27 (and I agree) that this may reflect confounding from temporal patterns in call composition. They refer to supplementary materials in the Stage 1 report for further detail which I unfortunately cannot access at the time of writing. However, "a similar pattern emerged for calls taken later during a work shift as indexed by call order". It is unclear to me why this measure would be similarly confounded as the authors state that "There is substantial variation in scheduling within individual nurses. Morning, afternoon, and evening shifts come in different configurations, with staggered start times and varying length". If shift start and finish times are highly varied, why would call order (i.e. within an individual nurses' shift) be confounded by temporal patterns in call type over the day? This is worth clarifying in the main text of the manuscript.*

Response: Thank you, this is a point well made. The main reason we see call order as potentially problematic in our setting (it need not be the case in other settings) is that it is correlated with time of day, and at the same time the variation in scheduling is not smooth over the whole day but rather concentrated inside 2-3 blocks of maybe 1-2 hours each (this can be seen quite well with the spikes for shift start times in Figure 1). Thus, on average, call order will be higher for calls taken later in the day, and in an unadjusted analysis, existing scheduling variation will not handle this completely. However, with additional modelling/adjustment, call order would probably work fine, for example we could have restricted analysis to cases where there would be "natural controls" (due to scheduling variation) such that higher call orders would match lower call orders conditional on patient type (time of day). This could probably be achieved by adjusting for time of day, shift starting time etc. in the analysis. We did consider using this approach for quite some time at the beginning of the project but we later abandoned it because we saw other better approaches available (overlap and breaks, which we eventually used instead). The main advantage with the overlap approach compared to using call order as described here is that it gives 1) a cleaner comparison and 2) a much wider difference in plausible fatigue, since we are comparing more like start to end of shifts, rather than – as would have been the case with call order – shorter time differentials between more vs less fatigue (more like 1-2 hours in that case).

We have now clarified this in the manuscript. In the Discussion section, fifth paragraph we edited/added the following text to highlight that the extent to which both time of day and call order may be problematic depends both on setting and overall analytic strategy: "For these reasons, care should be taken if using time of day or call order as proxies for fatigue when the goal is to estimate

influence on decisions. Both variables may induce confounding, but to what extent of course will depend on setting and overall analytic strategy. In our case, call order is problematic because it is correlated with time of day. Maybe this problem could have been alleviated by exploiting the staggered pattern of shift start times, but possibly not well enough, also considering there were other more suitable proxies for fatigue available in the data.”

We also included the supporting methodological discussion directly in the Stage 2 Supplementary Materials and we now refer to it here (as Supplementary Note 1) instead of, as previously, Stage 1 Supplementary Materials.

Comment 2. (2) *On p37 the authors state that “From a practical perspective these findings are reassuring. An implication for policymaking is that generic concerns about decision fatigue should have little influence on policy in healthcare or elsewhere.” I would recommend amending to acknowledge that this study raises additional questions with practical/policy relevance. The results make a persuasive case that decision fatigue may not be the mechanism which explains observed changes in health professional decision making over the working day. However, even if the patterns observed in the wider literature result from (suggested confounding) factors such as boredom or lack of motivation, this still has policy relevance.*

Response. We have done as suggested and added the following text at the end of that paragraph: “Also, policy may still have a role to play in addressing other, potentially related behavioural factors such as boredom or lack of motivation.”

Comment 3. (3) *On p37 the authors state that “Higher urgency ratings were assigned more often later in the day. It would have been easy to build a case (in good faith) where this pattern was taken as prima facie evidence of decision fatigue.... However, that interpretation would likely be wrong because the observed association between time of day and assigned urgency is likely confounded by other time of day dependent effects, for example composition of call types”. Some additional discussion at this point would be welcome as this will be a key point for the field going forward. There are examples of studies showing changes which align with decision fatigue from e.g. experimentally controlled studies where possible confounders are controlled (e.g. *Frontiers | Overcoming Therapeutic Inertia in Multiple Sclerosis Care: A Pilot Randomized Trial Applying the Traffic Light System in Medical Education*), from contexts where the decision in question/context is highly uniform (e.g. decisions to wash hands on entering a ward <https://pubmed.ncbi.nlm.nih.gov/25365728/>; scan reading, etc) and from contexts where the decisions under study are e.g. prescribing particular drugs for particular conditions (<https://journals.sagepub.com/doi/full/10.1177/0272989X241263823>) and where there is no obvious reason why one might expect that the number of patients presenting with osteoporosis, or deemed suitable for a statin for example would vary systematically across the day. This seems worth some additional discussion.*

Response. Thank you, we agree that the extent to which time of day may lead to confounding can depend on setting, e.g., it is particularly bad if correlated with patient composition when this is also expected to influence behavior directly, and at the same time no other measures are taken to handle this aspect in the analysis. But in some settings this may be less of a problem, as the Referee pointed out. We have now edited the end of this paragraph to better reflect that we indeed see time of day as a potential source of confounding but that the extent depends on setting. We added/edited the

following text (Discussion section, fifth paragraph): “For these reasons, care should be taken if using time of day or call order as proxies for fatigue when the goal is to estimate influence on decisions. Both variables may induce confounding, but to what extent of course will depend on setting and overall analytic strategy.”

Comment 4. (4) On p39 where the authors state “.....other organization-specific factors like training or guidelines might play a role”, it would be good to specify at this point whether decision support algorithms are used in this setting, and if they are used, the possible impact on the study results. In the comparable UK service, nurses utilise decision support software that scaffolds the decisions made. If such decision support systems are in place, this may have relevance to both the effort involved in decision making and to the variation in outcomes that is possible within each consultation.

Response. Thank you, yes there is such a system in place. It is symptom based and using it is part of the work process when handling calls. It would have been interesting to analyze variation in how this system is used both within and between nurses, but since we have no use data, we prefer not to speculate about possible effects in the paper. We now note in the paper that this system was available to nurses. In the Discussion section, eighth paragraph, we added the following text: “Also, nurses have access to a symptom-based, scaffolding decision support system. Using this system is part of the work process when handling calls. It would have been interesting to analyse variation in how this system is used both within and between nurses, but we did not have access to this data.”

Comment 5. (5) Finally, I would recommend aligning the phrasing of the abstract with the discussion/conclusion. In the discussion, the authors write that “the relationship between fatigue and information processing or cognitive effort avoidance could be more subtle or complicated than previously thought. It might be the case that there is a real effect that is more nuanced, weaker, potentially heterogeneous or context dependent”. In the conclusion, they write “Whereas these findings don’t preclude the existence of a weaker version of decision fatigue or more context specific effects, they cast serious doubt on the empirical relevance of decision fatigue as a domain general effect for sequential decisions in healthcare and elsewhere”. This is a measured way to summarise the results which encourages further investigation of possible mechanisms. It would be good if the abstract wording was consistent with this.

Response. We have done as suggested. We edited the final part of the abstract to better align it with the main conclusions of the paper. This part now reads: “Whereas these results don’t preclude the existence of a weaker or more nuanced version of decision fatigue or more context specific effects, they cast serious doubt on the empirical relevance of decision fatigue as a domain general effect for sequential decisions in healthcare and elsewhere.”

Comment 6. Minor point. p27: Please rephrase the sentence “Proportion consult ranged from 13.8% to 22.7%.” to make clear to the reader exactly what this refers to.

Response. Done. The sentence now reads: "The proportion of calls resulting in external consultation ranged from 13.8% to 22.7%."

Comment. *Thank you for the opportunity to review this paper. It's impressive work which has given me a great deal of food for thought.*

Response: Thank you once again for carefully evaluating our paper and for the helpful comments and suggestions during the revisions.